# Long Non-Coding RNAs and Their “Discrete” Contribution to IBD and Johne’s Disease—What Stands out in the Current Picture? A Comprehensive Review

**DOI:** 10.3390/ijms241713566

**Published:** 2023-09-01

**Authors:** Kostas A. Triantaphyllopoulos

**Affiliations:** Department of Biotechnology, School of Applied Biology and Biotechnology, Agricultural University of Athens, 75 Iera Odos St., 11855 Athens, Greece; ktrianta@aua.gr

**Keywords:** long non-coding RNAs, biomarker, epigenetic, regulation, mycobacter, Johne’s disease, Crohn, inflammatory bowel disease, colitis

## Abstract

Non-coding RNAs (ncRNA) have paved the way to new perspectives on the regulation of gene expression, not only in biology and medicine, but also in associated fields and technologies, ensuring advances in diagnostic means and therapeutic modalities. Critical in this multistep approach are the associations of long non-coding RNA (lncRNA) with diseases and their causal genes in their networks of interactions, gene enrichment and expression analysis, associated pathways, the monitoring of the involved genes and their functional roles during disease progression from one stage to another. Studies have shown that Johne’s Disease (JD), caused by Mycobacterium avium subspecies partuberculosis (MAP), shares common lncRNAs, clinical findings, and other molecular entities with Crohn’s Disease (CD). This has been a subject of vigorous investigation owing to the zoonotic nature of this condition, although results are still inconclusive. In this review, on one hand, the current knowledge of lncRNAs in cells is presented, focusing on the pathogenesis of gastrointestinal-related pathologies and MAP-related infections and, on the other hand, we attempt to dissect the associated genes and pathways involved. Furthermore, the recently characterized and novel lncRNAs share common pathologies with IBD and JD, including the expression, molecular networks, and dataset analysis results. These are also presented in an attempt to identify potential biomarkers pertinent to cattle and human disease phenotypes.

## 1. Introduction

### 1.1. Introduction to Intestine-Related Pathologies

Nowadays, serious concerns on the disease front have led to the annual publication of more than 10,000 papers worldwide in the field of infectious diseases, including those occurring in livestock and domestic animals. Importantly, in humans and animals, this amount increases each year [1]. Mycobacterial infections in the human population are responsible for nearly 1.5 × 10^6^ deaths, while 6.5 × 10^6^ individuals have been diagnosed with mycobacterial infections including those caused by the *Mycobacterium tuberculosis* (Mtb) pathogen, the causative agent of tuberculosis (TB) [2]. The causative agent Mtb adds to non-tuberculous (NTM) infections, which cause an estimated 1.7 billion people to be latently infected with Mtb, an example being the condition known as latent tuberculosis infection (LTBI), which does not have obvious disease symptoms [2] (also refer to Latent TB Infection and TB Disease, Centers for Disease Control and Prevention (CDC)).

The death toll figure associated with animal diseases is rising every year for domestic animals dying unaided. Although the condition of herds with Johne’s disease (JD) or paratuberculosis (PTB) has deteriorated worldwide, resulting in decreased milk production and increased cattle replacement costs [3,4,5], little progress has been made in terms of efficiently controlling the pathogen’s spread, which has inevitably led to enormous economic losses. As an example, the US dairy industry was estimated to cost USD 200 to USD 250 million annually, or USD 22 to USD 27 per cow, and this cost rises every year [6]. In more detail, *Mycobacterium avium* subsp. *paratuberculosis* (MAP), which is the causal pathogen of JD, is a chronic wasting ailment that occurs in ruminants and other species. In animal settings such as ruminants, including cattle infected by MAP, the latter pathogen is the causative agent according to Koch’s postulates [7,8], under which the symptoms and clinical findings manifest as JD.

Historically, MAP was first detected in milk samples from clinically symptomatic cows in 1929 [9]. Since then, further studies have proven the presence of MAP in milk from asymptomatic cows [10] as well as in colostrum [11]. In contrast to bovine milk, studies about the prevalence of MAP in dairy productive animals such as goats is very low. The pathogenesis in ruminants is characterized by the oral ingestion of MAP, followed by a long incubation period that enables the bacterium to survive within the host’s macrophage cells of the immune system. Intestinal epithelial cells (IECs) are the main port of entry for MAP, which preferentially invades the host through microfold (M) cells and enterocytes across the ileal and jejunal mucosa [12] (see Section 7.1). The spread of MAP, which mainly infects the intestinal tracts of animals and domestic livestock, is provoked by a long subclinical stage while the pathogen is shed into the environment with feces [13,14]. Additionally, younger animals and newborn calves may spread the disease either to the mothers through milking by carrying it from the ground or by transmitting the pathogen in a deadend circle from the infected mothers to uninfected newborns or by water and feed uptake; therefore, the detection of all animals at the subclinical stage is very important for JD disease control and the blockade of zoonotic transmission.

Importantly, the old type or current diagnostic methods (e.g., ELISA) are not suitable or adequate for the detection of subclinical animals that transmit the disease, and serum antibodies are usually not detectable in the preclinical phase. Therefore, the development of alternative diagnostic tools is urgently needed to control the disease early, ideally in the preclinical phase. More importantly, the detection of MAP in the feces via quantitative PCR is more reliable than the time-consuming fecal culture, in which the sensitivity in early subclinical infections is low and the cultivation of the Mycobacterium is too long (>3 months). The quantitative detection of MAP DNA in feces is performed by real-time PCR in compliance with the ISO17025 requirements [13,15], which also include the DNA isolation protocol, in which an 89 bp sequence region is amplified and detected within the MAP IS900 DNA element. Interestingly, a disease feature is that the clinical picture is variable with some MAP-infected animals presenting no clinical symptoms and others developing chronic intestinal inflammation that gives rise to severe weight loss and significantly decreased milk production, as it occurs in cattle and other ruminant species.

Initially, the infection is driven by the host’s cell-mediated immune response and incites a granulomatous inflammatory reaction in the intestinal tissue and mesenteric lymph nodes, resulting in protein-losing enteropathy, malabsorption, diarrhea, weight loss, and edema [16]. MAP infections initiate signaling pathways and several factors, including protective Th1 cell responses with INF-γ release, which leads to the activation of antimicrobial mechanisms in macrophages, phago-lysosomal maturation, acidification, and bacterial killing [17]. During the stage of infection, MAP makes an attempt to inhibit phagosomal maturation and acidification as well as to impair antigen presentation to T cells in order to enhance its intracellular persistence and survival. The latter is achieved by MAP through the modulation of IFN-γ signaling or by inducing increased secretion of IL-10 to promote bacterial persistence and establish infection [18,19]. MAP is resistant to difficult environmental conditions for long periods of time in the absence of host animals, which is evidenced by its detection in contaminated pastures and meadows that remains infective even after the transfer or removal of infected animals [20]. The pathogen is most commonly ingested orally, as MAP-contaminated dairy products may serve as a source of the earliest infections [21]. Furthermore, MAP infections depend on several factors including the infectious dose, the long incubation period of MAP, which is longer than 2 years in vivo, the protective Th1 cell response, and IFN-γ release followed by the activation of antimicrobial mechanisms; however, 10–15% of infected cattle will develop clinical signs of PTB [22], which results in MAP carriers not being detectable at the subclinical stage of the disease. Thus, as a result of the latter, MAP carriers remain untreated, transmitting the pathogen to the flock, the environment, or to other flocks. Inappropriate milking methods or storage of the milk can lead to contamination with MAP-containing feces or an introduction of the pathogen from outside the farm. Notably, MAP has been detected in bedding, water, and dust samples from cattle, sheep, and recently, goat farms in studies [23].

The resemblance of MAP-infected animals to humans with the disease resides in shared or similar signs/symptoms and clinical findings with PTB, which is known as inflammatory bowel disease (IBD), a refractory chronic inflammatory condition that affects the gastrointestinal tract (GI), having similar disease ontology characteristics. Macrophages exert an important role during IBD development. IBD-related pathologies, such as CD, share similarities in terms of histopathological and clinical findings with PTB [24,25,26,27,28,29]. Nevertheless, and despite the progress made on detection methods, the role of MAP in IBD, and especially CD, is still controversial due to conflicting evidence and a lack of reproducibility [29,30].

Two major subtypes of IBD have been reported, i.e., ulcerative colitis (UC) and CD [31]. As the occurrence of IBD is continuously increasing, particularly in developing countries, this disease has become a concern worldwide, as it does not have a proper cure and there are only symptoms-oriented interventions available. There are two main types of treatment for IBD: induction therapy and maintenance therapy. Currently, mucosal healing is regarded as a new therapeutic goal for reducing the rates of re-hospitalization, operation, and disability. Consequently, there is no form of cure for CD and no medication to ameliorate the disease substantially, and the chronic inflammatory disease symptoms remain in the gastrointestinal tract for life; although, trials that have examined the macrolide antibiotic effect have indicated that a cure for Crohn’s disease is possible [32]. Currently, the etiology of CD is still unknown, although the clinical findings in humans are similar to JD signs in ruminants, for example, in cattle affected by the MAP pathogen.

### 1.2. LncRNAs as Aides in the Disease Research

In recent years, incessant microbiological research has attracted the most powerful available resources to combat life-threatening infectious diseases, rendering this exploration significantly necessary and of high priority to assess the “quiver” that the pathogens carry and the susceptibility profile of the host. More importantly, new discoveries in the genetic and non-genetic components that have been identified in pathologies associated with bacterial invasion in human or animal cells have paved the way to advanced and in depth learning of the interactions between the molecular entities and environmental factors that are likely to lead to IBD-related pathogenicities. The ongoing research is significant and has strengthened the exploration and discovery of potential disease biomarkers that could improve diagnostic tools and the examination of new therapeutic modalities of high priority [33].

As mentioned earlier, accurate diagnosis and detection at the subclinical stage is a very important step for disease control. This is currently performed through the quantitative detection of MAP DNA by real-time PCR or sequencing methods in animals with the disease; the latter gives the potential to also detect regions in the coding and non-coding parts of the genome, as has been reported for microRNAs (miRNAs), as well as providing the ability to study infectious disease biomarkers [34].

In recent years, non-coding RNAs are a novel and very promising diagnostic approach to infectious and non-infectious diseases that has become a major focus of interest in research, and a large number of ncRNAs have been identified through large-scale genomic analyses [35,36,37]. Conventionally, ncRNAs can be broadly classified into small ncRNAs (maximum length of ≈200 nucleotides, including microRNAs, siRNAs, and piRNAs) and longer transcripts, including as lncRNAs (>200 nucleotides in length) and small RNAs (50 to 300 nucleotides long). Of the above categories, miRNAs have been widely studied in various diseases [38]. These functionally act by blocking at the post-transcriptional level or translationally repressing the target mRNAs. On the contrary, only a small number of long non-coding (lncRNAs) have been functionally characterized so far.

LncRNAs are newly discovered potential biomarkers, which have introduced us to novel perspectives on the regulation of gene expression, not only in biology and medicine but also in associated scientific fields and technologies. Importantly, the complex interactions and dynamic equilibrium between the genetic code and epigenetic signatures during disease progression is modulated by environmental signals over the course of time under a traceable path. LncRNAs and long intervening noncoding RNAs (lincRNAs) are known to act as decoys, scaffolds, sponges, and guides to proteins and RNA molecules in cells, fulfilling essential functions associated with gene expression regulation. Like miRNAs, this class of long ncRNAs has emerged as important regulators of both normal and pathological states, while compelling evidence has been accumulated recently, indicating that lncRNAs are involved in a wide range of biological functions [39]. Importantly, lncRNAs can function as tumor suppressors [40], oncogenes [41] during the development of multiple cancers, or mediators of infectious diseases by controlling the basal and *TLR2*-inducible expression of *TNFα* in human monocytes [42].

In this review, updated information and insights are provided in relation to recent discoveries in epigenetics and the interdisciplinary field of genomics by highlighting gene regulation by lncRNAs, which is supported by database searches, biotools, and literature mining for IBD-related pathologies, such as CD and UC in humans, as well as JD in ruminants, focusing on cattle which are vulnerable to MAP infection and JD disease. This comprehensive review is supported by the author’s database searches and analyses from which comparisons with the current state of knowledge on novel lncRNAs, uncategorized transcripts, and any new long non-coding genomic information are made. The collective presentation of all lncRNAs from the literature, including in silico analyses, is categorized as species-specific and disease-specific and is available in this review, and it will be a useful and frequently updated guide for future reference. Additions will be made should new potential biomarkers pertinent to associated intestine pathologies appear.

## 2. Morphology and Biological Characteristics of *Mycobacterium avium*

*Mycobacterium avium* is a species of the phylum *Actinobacteria*, belonging to the genus *Mycobacterium*. *Mycobacterium avium* subspecies *paratuberculosis* is a thin, straight or slightly bent, acid-resistant, and immobile bacterium. It is mainly aerobic, but under anaerobic conditions, it has shown signs of survival [43]. Its dimensions are normally 0.5 × 1.5 μm. MAP ribosomes are no different from typical bacterial ribosomes.

*Mycobacterium avium* is a rod prokaryote, zoonotic microorganism that causes avian tuberculosis and *Mycobacterium avium* complex (MAC) in humans. A secondary infection to AIDS, MAC, also called *Mycobacterium avium*-intracellulare complex, is a microbial complex of two Mycobacterium species, *Mycobacterium avium* and *Mycobacterium intracellulare*, which are saprotrophic organisms that are present in soil and water. Interestingly, disease-isolated mycobacteria from CD patients are identical to *Mycobacterium paratuberculosis*, as determined by DNA probes that can distinguish between mycobacterial species [32]. This advocates for MAP being the causal pathogen of CD in humans.

The genus *Mycobacterium* represents the only entity within the *Mycobacteriaceae* family, which belongs to the order *Mycobacteriales* and the phylum *Actinomycetota* [44]. Whereas the great majority of about 130 described species in the genus are harmless environmental saprophytes, some mycobacteria have evolved to be major pathogens. The pathogenic species mainly belong to the slowly growing mycobacteria category and comprise well-known human pathogens, such as *Mycobacterium tuberculosis*, *Mycobacterium leprae*, and *Mycobacterium ulcerans*, while the confirmed animal pathogens are *Mycobacterium bovis*, *Mycobacterium marinum*, and MAP [45]. The MAP pathogen has been implicated in JD and also has the potential to infect humans and be the causal pathogen of CD. Also, MAP has the characteristics of an acid-resistant pathogen mycobacteria, including the human pathogen *M. tuberculosis*, which is not a typical Gram-positive mycobacterium. Its lipid-rich cell wall makes it resistant to Gram staining; thus, it can only be stained with carbolfuchsin, acid–alcohol, and methylene blue [46]. In more detail, Ziehl–Neelsen staining for the direct detection of mycobacteria by microscopy is used to identify acid-fast bacilli, while the lipid-rich cell walls of mycobacteria make them resistant to Gram staining. MAP proliferates intracellularly with extremely slow growth in vitro, which lasts from 8 weeks to 4 months or more [47], while it requires the presence of iron and mycobactin (mycobactin—a hydroxamate iron chelatin) to support its growth [48]. In culture, the most commonly used nutrient for MAP is HERM (Herrold’s egg yolk medium agar). MAP can survive outside the animal for quite some time. In river waters, it can stay alive for 5 months and in lake waters, it can survive for 9 months [49]. In feces and soil, MAP can survive for 11 months, while in urine, it can only survive for 7 days [50]. Stools appear to act as bacteriostatic agents, and urine acts as a bactericide. In general, newborns and young animals are the most vulnerable groups in terms of infection by the MAP mycobacterium. In contrast, adult animals are considered to be resistant, although there is evidence that their infection is possible under certain conditions. Thus, the most frequent practices are to remove calves from their dams as soon as possible after birth and to keep the exposure of calves and heifers to adult cattle minimal [51]. MAP is transmitted by oral ingestion, mainly in the feces and, to a lesser extent, through contaminated milk and colostrum. The main source of infection for flocks is adult animals at an advanced stage of infection that shed the bacterium in large quantities [49]. In humans, *Mycobacterium* sp. can invade host cells and replicate inside, as many other bacterial pathogens, such as *listeriae* and *salmonellae*, do. Intrauterine transmission of JD disease seems to be possible only from pregnant females who are in an advanced stage of the disease. In fact, this mode of transmission was found in 25% of calf births by the mothers of sick animals with severe JD (PTB) [47].

## 3. MAP, Crohn’s Disease, and Relative Pathologies

Currently, there are divided opinions on the possible link between MAP and Crohn’s disease, albeit that MAP is involved in a constantly increasing number of pathologic conditions of unknown etiology in humans and is considered a potent threat to animals and human health [13,14,52]. More importantly, mycobacterial infections in livestock have become a matter of concern and a wide topic in animal research with multidisciplinary fields involved if we consider the wide range of species susceptible to MAP infections, which was emphasized in a recent report [53]. The available evidence covers a wide range of hosts including non-ruminant species such as pigs [54], rabbits [55], foxes and ferrets [56], and macaques [57], demonstrating MAP’s wide host infectivity and worldwide prevalence associated with the reduced productivity of infected animals [58].

More importantly, besides the threat to animal welfare, MAP is a controversial causative pathogen, since clinicians and scientists’ have divided opinions on the evidence of MAP isolation from human patients with Crohn’s disease, which was reformed recently by the positive MAP findings from patients [24,26,27,59]; thus, this pathogen poses a serious infectious threat to the public with its transmission occurring mainly through dairy products in livestock. The latter provides controversial evidence that MAP is zoonotic with a latent stage of infection similar to that of Mtb, the causative agent of TB in humans, where infection leads to a persistent immune response that controls but does not completely eliminate the pathogen. The latter makes JD difficult to document as a zoonotic disease that is associated with the pathogenesis of Crohn’s and related pathologies [24,26]; thus, the debate is still ongoing. The link of MAP zoonosis with CD has been a medical controversy for over one hundred years [27,60]; however, validation of the effort of Professor John Hermon-Taylor, an early elucidator of the zoonotic capacity of MAP [60], has come from numerous recent studies showing CD resolution with antimycobacterial therapies targeted against MAP [61,62,63,64,65].

Experimental evidence for MAP as the causative pathogen of IBD-related pathologies and CD has been reported in several studies in which the organism has been cultured from various tissues such as intestinal tissue, feces, breast milk, and/or blood from IBD patients and, recently, MAP DNA/RNA has been detected in patient samples versus healthy controls [25,27,28,59]. More importantly, exposure to MAP has not been accurately determined and the pathogen is easily spread by various means of contamination, posing a threat to the public, as the organism has been found in water, commercial milk, dairy products, etc., and has been shown to survive after pasteurization procedures that are sufficient to kill common contaminants [66]. This level of infection definitely leads to contamination of the environment via a contaminated water supply, dust bio-aerosols, milk, and the food supply. This contaminated environment not only affects the spread of MAP among animals, it may also be associated with the human intestinal disorders, i.e., IBD and CD [25,28,66,67]. Interestingly, MAP has been isolated from intestinal tissue, as well as from the peripheral blood of human patients suffering from a similar granulomatous inflammatory disease known as CD [25] Several pathogens, including MAP, have been claimed to be associated with CD in humans [26,32]. Reminiscent of JD, CD also affects the pediatric population. Thus, the identification of MAP in gut tissue and blood from pediatric IBD patients suggests the possible involvement of MAP in the early stages of CD development in children [49], while it was also suggested that CD is likely to be more than one disease, which complicates research efforts even more. Another study described the increasing presence of MAP in cattle (43%), buffalo (36%), goats (23%), and sheep (41%), while 30.8% of 28,291 humans (via serum ELISA, blood PCR, and stool PCR) tested positive for MAP [68]. Further to MAP’s involvement in these diseases, recent reports show evidence that MAP has been found within granulomas such as CD, but more importantly, it can stimulate autoantibodies in diseases such as type 1 diabetes (T1D) and Hashimoto’s thyroiditis, while beyond Crohn’s and T1D, MAP is increasingly being associated with host autoimmunity. Sechi and co-workers conducted several studies showing an association between MAP and T1D patients in Sardinia [69], which has the second highest incidence of T1D in the world [70]. Moreover, other reports also show evidence of MAP in T1D children [71,72] and have examined the genetic risk factor linking mycobacterial infection and T1D [73].

## 4. Long Non-Coding RNAs (LncRNAs) and Their Footprint in Gene Regulation

### 4.1. LncRNAs in the Disease State

The human genome is pervasively transcribed, while a small fraction only of all RNAs processed inside the cell are protein-coding sequences. In fact, only ca. 2% of the transcripts of the human genome can encode proteins, and less than 3% compose protein-coding gene exons [74]. Consequently, the majority of the transcripts seem to be non-protein-coding sequences, which account for 98% of the human genome (ncRNAs), although budding yeast has been proven to be a powerful model organism for understanding the mechanisms ruling pervasive transcription [75,76].

Known types of non-coding RNAs are transfer RNAs (tRNAs); ribosomal RNAs(rRNAs); small RNAs, such as microRNAs, siRNAs, piRNAs, snoRNAs, snRNAs, and exRNAs; long ncRNAs (lncRNAs); long intergenic non-coding RNA (lincRNA) lincRNAs; circular RNAs (circRNAs); and examples of ncRNAs, such as the well-known Xist and HOTAIR. circRNAs are stable, evolutionarily conserved, and single-stranded RNA molecules. Unlike linear RNAs, circRNAs are closed-loop type RNAs with joined 3′ and 5′ ends [77]. Four types of circRNAs have been discovered, namely exonic circRNAs (ecircRNAs), circular intronic RNAs (ciRNAs), exon–intron circRNAs (EIciRNAs), and intergenic circRNAs [78,79]. circRNAs function as miRNA sponges and can regulate RNA expression by consuming miRNA targets [80]. Additionally, circRNAs can interact with RNA-binding proteins to influence certain physiological processes [81] and can also act as gene transcription regulators [82].

It was shown that bacteria interfere with the expression of mammalian regulatory RNAs to modify immune signaling, autophagy, or the apoptotic machinery, and lncRNAs were reported to play a crucial role in the regulation of eukaryotic gene expression. This is in contrast to the known role of miRNAs in bacterial infections, which has been extensively studied and reviewed over the years [83,84,85]; thus, the emerging potential of RNA entities, collectively known as lncRNAs, has already begun to be manifested through their regulatory competence in complex machinery, illuminating the “usefulness” of read through pervasive transcription, i.e., a transcriptional read through of transcripts with non-established functions, leading to the accumulation of many opportunistic transcripts, whose synthesis is controlled by the cell at the co-transcriptional RNA processing level; this phenomenon can lead to chimeric transcripts and retained introns [86]. More importantly, it also demonstrates the dazzling array of opportunistic regulatory transcripts inside the cell, emphasizing the pivotal role of another class of regulatory RNA molecules that the cellular mechanism has available.

LncRNAs, among their many “charismatic” roles in gene activity, also play a critical role in organizing the 3D genome architecture and regulating *in cis-* or *in trans-* gene expression through numerous mechanisms that have been reported elsewhere [87,88,89,90]. They are DNA elements that can be encoded almost anywhere in the genome, e.g., within intergenic regions (lincRNAs), within protein-coding genes (in the antisense), and within introns [91]. Processed transcripts are a recent addition to this list. These are believed to express lncRNAs that can have important regulatory roles on their protein-coding counterparts [92].

The first non-coding gene was discovered in humans by using differential hybridization screens of cDNA libraries, a suitable approach to cloning and studying genes with tissue-specific and temporal patterns of gene expression. The aforementioned non-coding gene was the imprinted and maternally expressed transcript H19, which was initially classified as mRNA, but the absence of a long and conserved open reading frame (ORF) within *H19* and a lack of ribosomal interaction led to the conclusion that it is a non-coding transcript. Following studies led to the discovery of *XIST* in 1992 [93,94], which was then followed by the discovery of *AIRN*, a transcript that is responsible for the imprinting of the *IGFR2* gene in 2002 [95,96] and the multinetworked *MALAT1* transcript in 2003 [97]. Since 2006, numerous lncRNAs have been encoded within *HOX* gene clusters that regulate gene expression, similar to *HOTAIR* [98]. These studies revolutionized our perception of non-protein-coding gene functions and the biological relevance of lncRNAs in biology and medicine, while the identification of new lncRNAs has not only continued to increase but has impressively surpassed that of protein-coding transcripts [99].

Strictly speaking, for immune cells and the pertinently activated mechanisms upon invasion by bacteria or other microorganisms, the competing endogenous lncRNAs are associated with dynamic changes in gene expression, the products of which combat infectious pathogens, initiate repair mechanisms, and resolve inflammatory responses in cells and tissues. The primary structure of lncRNAs shows that are at least 200 nucleotides, although there is a diversity that distinguishes lncRNAs from smaller non-coding RNAs such as tRNA, miRNA, piRNA (Piwi-interacting RNAs), snRNA, snoRNA, etc. LncRNAs generally range in size from around 1 kb to longer than 100 kb, although they span an ORF containing a single exon, a minimum distinctive feature from other non-coding RNAs, mainly based on their size being larger than 200 ntds, which is useful for laboratory practices. LncRNAs are similar to protein-coding genes, but they are generally shorter, have fewer but longer exons, and possess low evolutionary conservation, which complicates the search for related domains and comparative investigations between species, despite them being located in highly conserved genomic regions [100].

As far as the intergenic lncRNAs (lincRNAs) are concerned, it was found that lincRNAs are transcriptionally activated similarly to mRNAs, as they are more conserved than introns and antisense transcripts, are more tissue-specifically expressed than protein-coding genes [101], and are more stable than intronic lncRNAs [102].

LncRNAs function as protein scaffolds, activators or inhibitors of transcription, and antisense RNA or miRNA sponges [103] that exhibit lower cellular concentrations than protein-coding genes but with a higher degree of tissue specificity [104], despite their low evolutionary conservation. Importantly, lncRNAs have a key role in the regulation of gene expression and, as mentioned earlier, remain poorly identified and annotated and are characterized less emphatically in domesticated animals compared to in other species such as humans and mice [105]; however, progress has been made in this decade, starting with bovine and pig genomes and including non-coding transcripts [106,107].

As shown in Table 1, at present, there are 96,411 lncRNA genes, and 173,112 lncRNA transcripts have been identified in human genome by next-generation sequencing, according to the NONCODE 2020 database (http://www.noncode.org, accessed 30 November 2022). In comparison, the current number of human protein-coding genes is 19,988 and there are coding transcripts 87,814, while the number of mouse protein-coding genes is shown in the GENCODE statistics (Table 1) (https://www.gencodegenes.org/human/stats.html).

The distributions of the protein-coding and non-coding information are demonstrated in Table 1 as percentages of each genetic element according to GENCODE and NONCODE statistics and the genomic information from annotation releases 110 and 106 for humans and cattle, respectively, as retrieved from the NCBI. Unlike the human genome the bovine genetic element information is not complete, and currently, there is an approximation of its distribution. Furthermore, the collected and presented ncRNA data shown in Table 1 use statistics from the NONCODE and GENCODE databases, originated from three main sources: (1) literature mining, (2) GenBank, and (3) specialized database such as Ensembl, RefSeq, lncRNAdb, lncipedia, LncRNADisease v2.0, etc. The other species, including cattle, use additional publication sources with updated genomic element profiles [108].

According to the human release statistics, the following descriptions explain the nature of lncRNAs and small ncRNAs (i.e., long non-coding RNA genes are the following gene biotypes: “processed_transcript”, “lincRNA”, “3prime_overlapping_ncrna”, “antisense”, “non_coding”, “sense_intronic”, “sense_overlapping”, “TEC”, “known_ncrna”, “macro_lncRNA”, “bidirectional_promoter_lncrna”, “lncRNA”. Moreover, small non-coding RNA genes are of the following gene biotypes: “snRNA”, “snoRNA”, “rRNA”, “Mt_tRNA”, “Mt_rRNA”, “misc_RNA”, “miRNA”, “ribozyme”, “sRNA”, “scaRNA”, “vaultRNA”). Non-coding transcripts are coarsely categorized using their biotypes and the following criteria: (a) well characterized—antisense, Mt_rRNA, Mt_tRNA, miRNA, rRNA, snRNA, snoRNA and (b) poorly characterized—3prime_overlapping_ncrna, lincRNA, misc_RNA, non_coding, processed_transcript, sense_intronic, sense_overlapping, as reported in GENCODE (GENCODE: https://www.gencodegenes.org/pages/biotypes.html, accessed 30 November 2022) [109].

### 4.2. Principles of Classification for Long Non-Coding RNAs

Strictly speaking, for lncRNA classes, lncRNAs can be arbitrarily categorized into four types, as shown in Figure 1: (1) **antisense lncRNAs** are transcribed in the opposite direction to protein-coding genes, and they often overlap by at least one exon, while they initiate transcription within or at the 3′ end of coding genes. The antisense lncRNAs initiate transcription from the aligned antiparallel protein-coding gene sequence and transcribe in the opposite direction to the overlapping coding exons, e.g., antisense *Tsix*, which negatively regulates *Xist* in *cis* via chromatin modifications at the onset of X-inactivation. (2) **Intronic lncRNAs**, initiate bidirectional expression inside an intron of a protein-coding gene and terminate expression without overlapping exons. (3) **Bidirectional lncRNAs** initiate transcription in a divergent fashion from a promoter of a protein-coding gene; the distance of lncRNA that constitutes the bidirectionality in transcription is not defined and is associated with imprecise termination, but is generally within a few hundred (<1000) base pairs. (4) **Intergenic lncRNAs** (which are often referred to as large intervening/intergenic non-coding RNAs or lincRNAs) are sequences of lncRNA that do not overlap with protein-coding genes as they have separate transcriptional units from protein-coding genes. Guttman et al. [110] and more recently Melé et al. defined the requirement of a lincRNA being located 5 kb away from protein-coding genes [111].

LncRNAs share many of their structural and functional features with protein-coding RNAs (mRNAs). The majority of the lncRNAs in mammals are produced by RNA-polymerase II (pol II), and many of them contain a 5′ cap, undergo post-translational modifications and splicing, and are polyadenylated at their 3′ ends [112]. In contrast, RNA-polymerase I (pol I) and RNA-polymerase III (pol III) are commonly limited to the transcription of housekeeping RNA transcripts [113,114]. Many lncRNAs exhibit alternative polyadenylation sites upstream of the 3′-most exon [112].

LncRNAs are conserved in a cell-type-specific manner and can vary in response to environmental stimuli and during development. They are poorly conserved in sequence, whereas the secondary structure of lncRNAs seems to be conserved across different species. When trying to assign individual functions to individual lncRNAs, one has to remember that a single 1000-base lncRNA is enough to carry out a large number of functions, possibly with different subsets of these functions (e.g., depending also on the presence in the lncRNA sequence of small sequence elements, such as, e.g., the *k*-base oligonucleotide-short motif), which can be active in different tissues and at different stages of development [115].

## 5. LncRNAs’ Involvement in Immune Dysregulation in IBD and JD

### 5.1. LncRNA Evidence in IBD-Related Pathologies

Recent evidence indicates that lncRNAs play important roles in the immune response, such as orchestrating the development of disparate types of immune cells, organizing chromatin and regulating the transcriptional programmes, and distinguished roles that feature immune homeostasis and gene activation. These are particularly important in infectious and inflammatory diseases, as perturbations in the non-coding component of the genome, such as RNA 3′ end processing and regulators of cleavage and polyadenylation, contribute to several human diseases [112]. LncRNAs also play important roles in directing the development of diverse immune cells and controlling the dynamic transcriptional programmes supported by alternative splicing and polyadenylation events that are hallmarks of immune cell activation. This activation of immune cells is associated with dynamic changes in gene expression, the products of which combat pathogens, initiate repair, and resolve inflammatory responses in cells and tissues [112].

Landmarks in the study of lncRNAs since their discovery 20 years ago include the significant progress in laboratory methods, computational biology, and algorithms invented to map new lncRNAs; however, the difficulty with accelerating progress resides in the low pace of characterization and the elucidation of their functional roles, which are mainly unexplored. Another important reason for their frequent presence in analyses in recent years is their implication in dozens of functional examples that have emerged involving numerous cellular processes ranging from embryonic stem cell studies to critical diseases like cancer or immune conditions, such as rheumatoid arthritis or infectious diseases [116].

One way to initiate the study of lncRNA characterization and functional analysis is by Gene Ontology (GO) term examination and enrichment analyses of the regional protein-coding genes [117,118]. The potential functions of the lncRNAs can be predicted by examining the Gene Ontology (GO) term enrichment of the nearest protein-coding gene. Additionally, for many tissues, the GO terms with highly significant enrichment are associated with relative functions of lncRNAs, lincRNAs, etc. that are essential for those tissues and have been examined and analyzed in previous studies of mammalian lncRNAs [111,119]. NFκB proteins are a family of five structurally related transcription factors that control the expression of inflammatory molecules, thereby counteracting bacterial infections. Many lncRNAs have been reported to interfere with NFκB signaling [120,121]. Specific examples of the latter interference account for (a) the induction of the lincRNA–Tnfaip3 complex, which is required for the transactivation of NFκB-regulated inflammatory genes in response to bacterial LPS stimulation [120] and (b) the reduction of *HULC* expression in a highly upregulated lncRNA (*HULC*) in liver cancer through treatment by TNF-a-induced apoptosis caused a reduction in HULC expression by the modulation of miRNA (miR-9) expression in association with DNA methyltransferases [122].

Importantly, lncRNAs involved in immune regulation or deregulation, such as *H19*, *DQ786243*, *BC012900*, *CDKN2B-AS1*, *NEAT1*, *CCAT1*, *MALAT1* (Table 2), and other listed transcripts that have been characterized and novel, inferred, experimentally discovered and functionally analyzed or uncategorized transcripts are involved in the regulation of gene expression in particular tissues (or ubiquitously expressed) or can be associated with a disease; thus, it is very likely that the condition is pathologically related to the particular tissue of origin where lncRNAs are expressed. For instance, GO terms related to the circulatory system are prevalent in tissues with a high density of blood vessels. Likewise, the aforementioned lncRNAs are linked to the intestinal epithelial cells, the innate immune responses, inflammation, and diseases of the gut such as, IBD, UC, and CD, according to published reports [122,123,124,125,126,127].

Despite the fact that a plethora of studies on IBD have focused on protein-coding genes, new advances in the field of lncRNAs, have shown their implications and altered expression in IBD patients compared to healthy controls. Although lncRNAs are novel gene regulators that have not been explored as much as miRNAs, an increasing number of lncRNAs are being implicated in IBD pathogeneses. Therefore, in Table 2, the author has collected published lncRNA information as well as information from several platforms and database searches of the human genome. An updated list of lncRNAs that are currently implicated in IBD-related pathologies list is presented in a concise form, where possible. In more detail, the results of the list shown in Table 2 include information from previous reports [123,127,128,129,130,131,132,133,134,135,136,137,138,139,140,141,142,143,144,145,146,151,152] supplemented by sources from the updated build version of the human genome (assembly GRCh38/hg38), including non-coding and coding genome databases such as NONECODE, GENCODE, NCBI, Ensembl, UCSC genome browser–Gene Interactions, GeneCards/GeneHancer/GeneTargets, Ensembl, GeneCards, and the gene profiler *Elixir* platform [151,152,153,154,155]. A sample of ncRNA-oriented databases/biotools involved in this research includes RNAcentral, LNCipedia, the GWAS Catalog, LncRNADisease v2.0, starBase v2.0, EVLncRNAs2.0, lncRNAfunc, LncRNAWiki 2.0, etc. Therefore, the human lncRNA list in Table 2 presents established as well as novel, uncategorized, and/or uncharacterized lncRNAs according to patients’ diagnoses in CD, UC, IBD, or experimental colitis model groups. This is anticipated to be a constantly updated depository for reference that can support further research on IBD-related pathologies. Moreover, the novel, uncharacterized/uncategorized, or Ensembl processed transcripts, mainly with unknown functional roles, are highlighted using bold fonts, while the information presented in Table 2 will be regularly updated by relevant databases, such as GWAS Catalog [156,157].

Although lncRNAs and the new class of circRNAs have not been thoroughly explored in relation to IBD, these RNAs seems to play crucial roles in tumor angiogenesis and carcinogenesis, since lncRNAs and circRNAs have recently been significantly identified as biomarkers for tumor diagnosis and prognosis [158]. Here, we present some examples from Table 2 to demonstrate the multiple roles that the novel *NCRNA00194* (GeneCards: *NRON*, gene) gene possesses. This gene is located on chromosome 9 and is associated with IBD but has also other functional associations, such as AIDS, Down syndrome, and heart disease [145]. Another example is the *LINC00928* transcript that is located on chromosome 15 and has functional associations with COPD, hypertension, and multiple myeloma (GWAS Catalog, EMBL-EBI), e.g., the variant and risk allele for hypertension rs28792763-A [159]. Furthermore, studies on the peripheral blood macrophages (PBMCs) of patients with UC have shown associations with other ailments, including IBD [160] and CD [161], for the long intergenic ncRNA *LINC01882* in chromosome 18 (Gene ID: *ENSG00000266961*) with the variant risk alleles rs2847278 (*p*-value 8 × 10^−28^) and rs2542151-G (*p*-value, 5 × 10^−17^), respectively (as appears in the GWAS Catalog). Also, Table 3 (lncRNAs in JD) was created from the analysis, based on searches by keywords of the disease type or sets of lncRNAs that have been reported in the literature. These were processed and analyzed using the aforementioned platforms and algorithms such as the Database Resources of the National Genomics Data Center, China National Center for Bioinformation in 2022, EVLncRNAs2.0, and data from NONCODE, EVLncRNAs2.0, LNCipedia, and LncRNADisease v2.0.

### 5.2. LncRNA Evidence from JD-Related Pathologies

Naturally infected cattle by MAP, studies have shown to contain lncRNAs with fewer exons than mRNAs (some of them incorporate less than three exons and some just have a single exon), while their length is also shorter than the length of mRNAs [105,162,163]. In more detail, Marete at al. published a report on the gene expression of highly differentially expressed gene targets associated with their affiliated lncRNA sequences and showed a pervasive lncRNA distribution on bovine autosomes (1–29) and on the X chromosome, which were experimentally localized in bovine chromosomes from macrophage cells of cows positive for JD vs. cows negative for JD (control) [163]. In a recent investigation by Gupta et al., 19 uncharacterized lncRNAs analyzed by RNA-Seq were shown to be differentially expressed (ten upregulated and nine downregulated lncRNAs) in bovine macrophages and in response to MAP infection [162]. The top upregulated genes in Marete et al.’s study, such as *FLRT2*, bta-mir-2285e-2 on chromosome 9, and 10 on the previous author’s list, were further away than five kilobases (kb), either upstream or downstream of their nearby genes (according to the classification based on their distance limits from the gene targets and their categories) and some were unknown or novel [163]. A gene enrichment analysis of the mRNA genes neighboring the lncRNAs involved revealed pathways related to immune response, such as coding genes that play roles in NF-κB2 signaling (i.e., TNFAIP3 interacting protein 3 (TNIP3), TNF-alpha-induced protein 3 (TNFAIP3), and NF-κB2). Additionally, more pathways and functions were implicated, such as organelle fission (MX1, MX2) and cytokine production (MAF, NFAT5) [163].

The lncRNAs identified in the Gupta et al. study were mostly intergenic (45.05%) with an average length of 600 bp long and ranged between 200 and 1000 bp [162], supporting the evidence that lncRNAs are mainly intergenic with a smaller overlap within genic regions [101]. The top 10 most significant differentially expressed (DEG) genes proximal to their lncRNA and a large amount of the above lncRNAs were found to be transcribed on chromosomes 18 and 19. These chromosomes are associated with signaling pathways for innate immune response, such as TNF signaling pathways (CCL5, CCL2, MAP2K3, MAP2K4, MAP3K14, MLKL, PIK3R5 and SOCS3), chemokine signaling pathways (CCL5, NFKBIB, ARRB2, CCL2, CCL4, CCL8, PIK3R5), NOD-like receptor signaling pathways (CCL5, NFKBIB, NLRP1, CCL2) and cytokine–cytokine receptor interactions (CCL5, CCL2, CCL4, CCL8, CSF3) [162]. Also, the identified lncRNAs are worth studying as they may contribute to the regulation of IFN signaling during MAP infection.

RNA-seq studies in bovine macrophages infected with MA revealed a significant log2 fold-change for the DEG. The frizzled class receptor 1 gene (*FZD1*) was DGE, and its fold change was estimated to be 15.56 times more highly expressed in macrophages for positive vs. negative for JD macrophages in cattle [163]. In addition, a second significant enriched pathway was the RNA polymerase I promoter escape, which involves the *TWISTNB* gene and encodes RNA polymerase I subunit F. It is upregulated in macrophages from JD-diseased cows. More importantly, the longest significant lncRNA that was found to be more highly expressed in macrophages from JD-diseased cows than in JD non-infected macrophages was located in the region adjacent to the FZD1-encoded transmembrane protein and associated with the cellular component of focal adhesion, which might explain the phenotypes observed for macrophages of JD [163].

Combining the reference search and ncRNA-dedicated databases/biotools, the results are shown in Table 3 below, including important established, novel, and uncategorized lncRNAs associated with bovine JD and/or mycobacterial disease. These can be used as a further reference and for research. Interestingly, Opa interacting protein 5 antisense transcript (*OIP5-AS1*) is the single lncRNA that was also enriched in cattle JD pathology, reminiscent of the findings of the analysis performed with the EVLncRNAs2.0 platform (*OIP5*, *ENSBTAG00000010766*) for the lncRNAs associated with bovine JD and/or mycobacterial disease (Table 3). The latter lncRNA is a conserved gene that maintains cell proliferation in embryonic stem cells and binds to and negatively regulates the activity of multiple cellular RNAs and microRNAs, including cyclin G associated kinase and ELAV-like RNA binding protein 1 [164]. In humans, *OIP5-AS1* is associated with several pathologies, including cancers such as gastric cancer as well as cancers and neoplasms of the colon, colorectal cancer, etc. It is supported by the Malacards database with cited work (https://www.malacards.org) [165]. In general, the *OIP5-AS1* transcript is a multifaceted lncRNA that occurs in humans and animals and has regulatory functions in normal cellular processes as well as in the development and progression of numerous disease states [166].

### 5.3. Differences in LncRNA Profiles of CD and JD

Summarizing the above results, the presentation of the retrieved lncRNA information, which is shown in Table 2 and Table 3, provides detailed lists of characterized and categorized lncRNAs, including novel transcripts from human IBD-related pathologies as well as bovine infectious diseases (mycobacterioses) of the gastrointestinal system. As mentioned earlier, Table 2 and Table 3 incorporate information such as comprehensive gene-centric lists that were retrieved from lncRNA databases and the protein-coding sequence and general nucleotide databases mentioned earlier in Section 5.1. Additionally, the usage of updated versions of more ncRNA-oriented databases/biotools that were not cited earlier, such as LncRNA2Target versions V2.0 and V3.0, Open Targets Genetics v22.10, The Human Reference Protein Interactome Mapping Project (HuRI), lncRNAfunc, LncRNADisease v2.0, starBase v2.0, EVLncRNAs2.0, and the updated LNCipedia_5.2, was essential in this study [167,168,169,170,171,172,173,174]. Moreover, LncRNAWiki 2.0 was used. This is a full-featured platform for ncRNA functional annotation as well as a tool for the network analysis created under the LncACTdb 3.0 platform [155]. An important part of the information for the lncRNAs shown in Table 2 and Table 3—analyzed from the EVLncRNAs 2.0 database—is the target gene interactions between lncRNAs and important coding and non-coding transcripts, as shown in Appendix A, which are regulated by lncRNAs. They are well-known to be molecules with regulatory roles in chromatin remodeling. Details in Table 3 for cattle include chromosome locations, exons, interaction types, targets, NCBI accession numbers, and descriptions supported by citation IDs (PMIDs). More importantly, the subcellular localization of particular lncRNAs provides important insights into biogenesis and functions and examples of cytosolic lncRNAs, such as *DANCR* and *OIP5-AS1*, are shown in Table 3 with details about their interactions and functions or other features, such as nuclear lncRNA NEAT1 (also shown in Table 2), which localizes to paraspeckles in the nuclear domains [175].

A notable example is a variation in the genetic locus of protein tyrosine phosphatase 2 (PTPN2) in UC and IBD (Table 2), which regulates cytokine signaling by acting on multiple phosphorylated proteins [147]. A study of patients with CD demonstrated a link between the SNP rs2542151 and lower levels of the PTPN2 protein in colonic fibroblasts, as well as the formation of aberrant autophagosomes in intestinal epithelial cells (IECs) [152]. The *PTPN2* locus SNP rs2542151 is related to the variation in lncRNA *LINC01882* (Table 2), which is primarily expressed in T cells and is also involved in IL-2 expression, affecting important events such as differentiation, immune responses, and the homeostasis of lymphocytes, including Tregs mechanisms. Specifically, the transcript *LINC01882* has been reported to play significant roles in autoimmune diseases, including IBD, and in peripheral blood mononuclear cells (PBMCs) of UC patients [176]. Additionally, the lncRNA *ROCKI* (Table 2) negatively regulates its cognate encoding gene, myristoylated alanine-rich protein kinase C (MARCKS), which promotes inflammatory cytokine and chemokine production. Thus, the above consecutive events show that MARCKS’ gene expression, mediated by *ROCKI*, contributes to the IBD pathology [177]. More importantly, the genetic variants that affect *ROCKI* expression have been linked to reduced risks of certain inflammatory and infectious diseases in humans, including IBD and TB. The author’s network analysis under the LncACTdb 3.0 database predicts a role of *ROCKI* in bovine tuberculosis (transmitted by *Mycobacterium bovis*) and its neighboring nodes in the gene network of lncRNA *DANCR* and miR-335-5p (Appendix A).

Consistent with the above and the contribution of lncRNAs in IBD, which has been also shown in several studies, is their expression profile which can successfully distinguish IBD patients from healthy controls [178]. Furthermore, the transcription characteristics and clinically relevant parameters of lncRNAs indicate that they have strong potential to be used as prognostic biomarkers in IBD [179]. Moreover, differences in expression levels have been also found for various novel lncRNAs (e.g., *KIF9-AS1*, *LINC01272*, and *DIO3OS*) in tissue and plasma samples of patients with IBD, rendering them as potential biomarkers of therapeutic or diagnostic value for IBD [180]. In the latter study by Wang et al., the Receiver Operating Characteristic Curve analysis (ROC) was employed to determine the specificity and sensitivity of these lncRNAs as potential diagnostic biomarkers for IBD-related pathologies, and the value of lncRNAs, *KIF9-AS1*, *LINC01272*, and *DIO3OS* as novel biomarkers for IBD was shown [180].

Also, the lncRNAs included in Table 2 are *GUSBP2*, *GAS5-AS1*, *LINC01272*, *DDX11-AS1*, *IFNG-AS1*, *KIF9-AS1*, and *BC012900*, which are specifically upregulated in IBD-related pathologies [125,181], while others, such as *ALOX12P2*, *DPP10-AS1*, *DIO3OS*, *lnc-PTPN2-2*, *TRIM52*, *TALAM1*, and *MALAT1*, are downregulated in plasma samples from patients with CD [125,176,180,182,183]. Interestingly, *ANRIL* (or *CDKN2B-AS1*) (Table 2) is one of the most downregulated lncRNAs in CD, and its transcript has also shown decreased levels of expression in ulcerative colitis (UC) colon tissues in recent studies [126,148]. Ultimately, a small number of lncRNAs show alternate behaviors (e.g., *NEAT1*) for either the type of tissue or the pathology, e.g., in serum versus colonic tissue of the DSS-induced colitis experimental model, as shown in Table 2 [123,184]. Additionally, lncRNAs involved in CD and datasets for UC and IBD were found in the author’s dataset analysis of CD patients from certain datasets (GEO, GSE75459). This information is included in Table 4 shown in Section 5.4.

### 5.4. Expression Profiling in Crohn’s Disease vs. Healthy Controls

In this section, an analysis of datasets assigned with GEO accession number GSE75459 under platform GPL16956 from the Gene Expression Omnibus (GEO)/NCBI was performed. In more detail, the Platform GPL16956 Agilent-045997 Arraystar human lncRNA microarray V3 (Agilent Technologies Inc., Santa Clara, CA, USA) contains several series of samples from various human disease conditions, including (the GSE75459 dataset) CD samples under the title name “Plasma Long Non-coding RNA and mRNA Expression Profile of Crohn’s Disease identified by Microarray”. The latter dataset analysis yielded novel lncRNAs and mRNAs for gene expression profiling with new targets in CD through a genome-wide analysis. The limma package was applied for this analysis, which is the core of the underlying computational engine of the GEO2R (http://www.ncbi.nlm.nih.gov/geo/geo2r). This generated up to the 250 most statistically significant differentially expressed genes by calculating an adjusted p-value using the Benjamini–Hochberg method to control the false discovery rate (FDR) [185]. In more detail, the results from the CD microarray experiment detailed above produced a list of non-coding RNAs, which are also included in Table 4 for the analysis of the serum samples (GSE75459) from CD patients versus normal (control) samples.

Thus, the results from the GSE75459 dataset analysis (Appendix A) include the upregulated genes at the top and the downregulated genes at the bottom (microarray analysis column, Table 4) in terms of differentially expressed long non-coding RNA genes. In Appendix A, gene expression is ranked in descending order for values of log2(fold change) or log2(FC), according to the directions of gene expression changes. Appendix A consists of seven columns, i.e., the sequence name, type of transcript, the log2(FC), the −log10(*p* value), the accession number of the sequence(s), the accession length (bp), and a brief description of the gene/transcripts. There are novel and uncategorized lncRNA transcripts for CD or common gene transcripts in Appendix A, which share transcripts that are also included in Table 2 for CD, UC, and IBD. Common lncRNA matches are *MALAT1* var3, 2, and 1 and *GAS5-AS1* (Appendix A), while certain variants of lncRNAs, such as *ALOX12P2* var2, *ALOX12P2* var1, *ALOX12E*, *TRIM52* var2, *TRIM52* var3, *TRIM52* var1, and *TALAM1*, are novel transcripts (results from the analysis of GSE75459 dataset (Appendix A)).

The LncRNAs *GUSBP2*, *GAS5-AS1*, *LINC01272*, *DDX11-AS1*, *IFNG-AS1*, *KIF9-AS1*, and *BC012900* (Table 2 and Table 4) are specifically upregulated in IBD-related pathologies [125], while others, such as *ALOX12P2*, *DPP10-AS1*, *DIO3OS*, *lnc-PTPN2-2*, and *TRIM52* (Appendix A) are novel transcripts. Furthermore, in the GSE75459 dataset analysis, the lncRNAs *TALAM1* and *MALAT1* were shown to be downregulated in plasma samples from patients with CD [125,176,180,183,186]. Interestingly, *ANRIL* (*CDKN2B-AS1*) (Table 2 and Table 4) is one of the top candidates and is the most downregulated lncRNA in CD, and its transcript also shows decreased expression in UC colon tissues [126,148]. Ultimately, a small number of lncRNAs show alternate behaviors (e.g., *NEAT1*), either for the type of tissue or the pathology, e.g., in serum versus colonic tissue of the DSS-induced colitis experimental model as shown in Table 2 and Table 4 [123,184]. Interestingly, in comparison with IBD, the genes associated with UC (Table 2) differ from the lncRNAs *CDKN2B-AS1*, *H19*, and *MALAT1* [152], with the common gene being the human lncRNA IFNG-AS1, which exerts its action in the IBD and/or UC subtypes.

Apart from the common transcripts/genes (the intersecting gene subset in both disease networks, i.e., IBD vs. JD), there are also unique molecules that are not found in databases or reported elsewhere, such as uncharacterized loci from the GSE75459 dataset analysis, e.g., XLOC and TCONS (Appendix A) and the newly reported RNA genes, such as *FIGNL2* Divergent Transcript), *ALOX12P2*, *TALAM1*, and others, as described below. Importantly, lncRNAs that are not common in CD and UC are GUSBP 16, 3, 15, and 14, *ALOX12P2* var2, *ALOX12P2* var1, *ALOX12E*, *FIGNL2-DT*, and *TALAM1*. Furthermore, the ncRNA transcripts GUSBP, FIGNL2-DT, GOLGA2P8, and lncAB107.3, failed for annotation under several programs (e.g., KOBAS, ECORI in StarBase etc.). It is assumed that these are novel transcripts that are not included in databases or that databases have not been updated yet.

The author employed various tools (PhyloCSF, LNCipedia, etc.) to assess the coding potential of the novel *GUSBP*, *ALOX12P2* var2, *ALOX12P2* var1, *ALOX12E*, *FIGNL2-DT*, *TALAM1*, and *GOLGA2P8* transcripts which have been reported elsewhere [171,187,188,189]. An interesting case is *MALAT 1* (metastasis associated lung adenocarcinoma transcript (1), also known as *NEAT2* (non-coding nuclear-enriched abundant transcript (2) and its natural antisense transcript (NAT), while the latter was identified in the *MALAT 1* locus, contributing to the stability of *MALAT1* by promoting the 3’ end cleavage and maturation of *MALAT1*. Importantly, *MALAT1* was found to be downregulated in the intestinal mucosa of patients with CD and mice with (DSS-induced) experimental colitis, while it maintained mucosal homeostasis in CD through the miR-146b-5p-CLDN11-NUMB pathway [186]. Concurrently, lncRNA *MALAT1* positively regulates the transcription and transcript stability of *TALAM1*—which is an antisense affiliated transcript of its *MALAT1* partner—and therefore establishes a positive regulatory feedback loop at the *MALAT1* locus to attain high cellular levels of *MALAT1* [183].

Thus, Table 4 shows the phenotypic profiles of the expressed lncRNAs involved in human IBD pathologies as well as cattle JD and is a concise list of the results of the combined literature and database search, including the analysis of the CD dataset (GSE75459) presented in Appendix A and the potential applications as diagnostic or therapeutic biomarkers.

In conclusion, the more lncRNAs with profoundly altered expression are presented in Table 4. Owing to their roles in the pertinent pathologies and their citation scores, they are introduced as potential biomarkers for either human or animal disorders of the intestinal epithelium. Some of these lncRNAs can serve as potential biomarkers for the clinical evaluation of patients with IBD, while others are inferred from the in silico analysis of the CD dataset (GSE75459). Many of these lncRNA transcripts are considered to be potential candidate biomarkers due to the fact that their gene levels are differentially expressed depending on the severity and progress of the disease, reflecting the changes in their transcript levels, which can be further validated to ensure the monitoring of the above pathologies.

## 6. The Epigenetic Role of the LncRNAs Involved in Human IBD-Related Pathologies and Mycobacterial Infections of the Host

In recent years, we have reformed our knowledge and insight to comprehend the value of epigenetic contributions to health and disease and the roles of non-genetic alterations at the DNA/RNA sequence or molecular level. Epigenetic processes generate the epigenome, and these involve a variety of modulations and chemical modifications that do not affect the genetic code as such but instead include, and are not limited to, DNA methylation, histone modifications, chromatin remodeling, the regulation of gene expression by non-coding RNAs, genome instability, and any other force that contributes to the animal phenotype [190,191,192]. Thus, considering the fundamental role of epigenetic regulation in immunity, cells, and mechanisms and focusing on T cell immunity and T helper (Th) cells, we recognize the major role they play in the complex regulation of mucosal immunity. Epigenetic regulation of Th cells is significantly involved in the maintenance of homeostasis and mucosal immunity compromises, leading to failures, like those that occur in IBD pathogenesis or threatening infections in JD, triggering chronic inflammatory diseases, autoimmunity, and autoinflammation.

The International Human Epigenome Consortium and the Human Epigenome Projects have been initiated to understand and pass on knowledge on the overall epigenetic mechanisms involved in human health and disease [193]. The changes in non-coding RNA are believed to cause obesity, diabetes, and neurodegenerative diseases, affecting the lungs, liver, or other organs. Thus, the interactive roles of the microbiome and epigenetic regulation in human health are very important in intestinal dysfunction and IBD-related pathologies. More than 200 genes have been shown to influence susceptibility to IBD and related pathologies, most of which are involved in immune responses [194]. This is a complex network involving innate and adaptive immunity that protects the host from infectious diseases and cancer.

Genome-wide association studies (GWAS) and exome sequencing have identified that only ~7% of disease-associated genetic variations (e.g., single nucleotide polymorphism, SNPs) identified-to-date are localized to protein-coding genes [195]. The vast majority of genetic disorders caused by sequence alterations (e.g., disease-associated SNPs) are localized to the non-coding regions of the genome, including genomic loci expressing lncRNAs [196]. More importantly, the majority of identified disease SNPs fail to explain the variance in the studies, as the genetic component of most diseases is complex and difficult to determine. For example, epigenetic modulations exerted by environmental factors influencing ncRNAs, DNA methylation, and histone modifications may partly explain the missing heritability and inconsistent variance shown in some studies, especially in terms of psychiatric illnesses [197,198].

In recent years, we have acknowledged the actions of microbial pathogens like *Mycobacterium* sp. or viruses that regulate epigenetic processes to evade host immunity and cause diseases [199,200]. As mentioned earlier (Section 7), the importance of the lncRNAs’ contribution to IBD is their expression profile, which can successfully distinguish IBD patients from healthy controls, and has the potential to identify diagnostic and prognostic biomarkers in IBD [178]. In terms of the observed differences in the expression levels of lncRNAs that have also been found in IBD (e.g., *DPP10-AS1*, *KIF9-AS1*, *LINC01272*, and *DIO3OS*), among the most important features that are encompassed in their potential as biomarkers, there are their gene regulatory mechanisms as well as their epigenetic modulation potential, which enables their variable expression in multiple disease states. The latter modulatory feature is based on the methylation status of the lncRNA sequences, which often affects an lncRNA’s capacity to “sponge” miRNAs, regulating histone modification via their target miRNAs, and therefore, its capacity to form regulatory axes and vice versa, i.e., to also be regulated through histone modifications, such as methylation as well as HDAC overexpression [190].

The known DNA demethylase, (ten–eleven translocation protein that acts as an eraser protein in humans) (TET2) binds to the promoter region of the lncRNA *CDKN2B-AS1* (or *ANRIL*) and regulates its expression and downstream genes. Interestingly, overexpression of the TET2 protein inhibits the abundance of *CDKN2B-AS1*, resulting in a decreased risk of gastric cancer [149]. Interestingly, *PTPRE-AS1* is directly bound to WD repeat-containing protein 5 (WDR5), modulating H3K4me3 of the PTPRE promoter to regulate PTPRE-dependent signaling during M2 macrophage activation (alternatively activating the macrophage lineage, as described in Section 7.3) [201].

In MAP-infected cattle, *Mycobacterium* sp. is capable of reprogramming the host cellular machinery, leading to the evasion of both the innate and adaptive immune responses, followed by the establishment of infection and dissemination. Lessons can be learnt from *Mycobacterium tuberculosis* infection in humans as well as infections in other mycobacterioses, e.g., *M. leprae*, as these pathogens reprogram the epigenetic mechanisms of the host epigenome by inducing changes in histone modification, DNA methylation, and non-coding RNA molecule expression or even reprogramming the cell potency [199].

The author studied the regulatory relationship between human lncRNA and DNA methylation related in a disease-centric fashion to ulcerative colitis, as shown in Appendix A. After searching Lnc2Meth database [202], the differentially methylated transcripts in the selected disease (UC) are the transcripts *DAPP1-001*, *DAPP1-002*, and *MALAT1*. The Lnc2Meth database was employed for a DNA methylation study related to the lncRNA loci in UC, which is a manually curated database of regulatory relationships between lncRNAs and DNA methylation associated with human disease that was created using datasets from a microarray analysis (HM450k, Illumina Infinium 450k DNA methylation array) and WGBS (WGBS: whole-genome bisulfite sequencing); thus, our analysis was based on criteria such as methylation sites or different element types of the gene and by using search queries by transcript (transcript-centric) or disease (disease-centric) with a certain reference to methylation (values of 0.3 or above were selected) [202].

The results of the differentially methylated transcripts presented in Appendix A show that the *DAPP1-001* and *DAPP1-002* transcripts were found to be hypomethylated in coding and non-coding regions (i.e., 5′UTR, 1st exon and 200 bp downstream of TSS) of the lncRNA gene in *-cis* (cis-methylated lncRNAs). These results were produced not only in UC but also in related gastrointestinal (GI) tract malignancies and in different organs, e.g., colon adenoma, colon adenocarcinoma, hepatocellular carcinoma, etc. (Appendix A). Further discussing these results, we believe that DAPP1 may act as a B-cell-associated adapter that regulates B-cell antigen receptor (BCR) signaling downstream of phosphoinositide 3-kinases (PI3K) (Table 2). The latter family of enzymes are dynamic regulators of physiological and cellular processes and are involved in cellular functions such as cell growth, proliferation, differentiation, motility, the survival of intracellular trafficking, metabolism, and cancer. A recent study revealed that the 3-CpG methylation signature for a combination of three selected genes, including *DAPP1*, which was used to construct a prognostic risk-score model, was evaluated in the clinic as a prognostic biomarker for individualized survival predictions in colorectal cancer patients [203].

## 7. The Triarchy of Infection, lncRNA Intervention, and Regulation (IIR)

### 7.1. The Infection Stage

Considering, on one hand, the infectious route of MAP pathogen invasion and the “immune disturbance” of the host and, on the other hand, the “breach” of intestinal mucosal homeostasis and barrier dysfunction caused by IBD in humans, the following events represent a common ground for these pathologies: either (a) starting with the steps of an immune challenge, triggered by MAP, and followed by antigen presentation from the host macrophages or (b) a gradual rising chronic intestinal barrier dysfunction of an unknown trigger that is lurking and deteriorating and is manifested by the aforementioned associated signaling pathways in the previous sections. As an “unknown trigger”, the contemporary scenarios are considered and briefly described in this and the following subsections.

Evidence has accumulated through research over the years to show the interplay of genetic susceptibility, including non-coding regulatory transcripts, the environmental impact on the microbiome, and the immune condition of the host leading to or triggered by dysbiosis, which contributes to intestinal barrier dysfunction and inappropriate intestinal immune activation, thus resulting in disturbed intestinal mucosal homeostasis, which has become the hallmark of IBD [204,205]. The pathogen (MAP) starts its transmission from the oral mucosa and spreads to the ileum, which has been implicated as the primary point of MAP invasion upon the entrance of MAP into the microfold (M) cells of Peyer’s patches in the intestinal mucosa (Figure 2). M cells are highly specialized cells that are present within the intestinal epithelium overlying organized lymphoid follicles of the small and large intestines, which play the role of antigen carriers [206]. Thus, the M cells play a central role in the initiation of mucosal immune responses and surveillance by transporting antigens and microorganisms to the underlying lymphoid tissue, which actively transports external antigens from the gut lumen to the lymphoid follicles [207].

Studies on *M. tuberculosis* (Mbt) infection provide evidence that bacterial toxins of lipopolysaccharides (LPS), mycolic acid, and the lipoarabinomannan (LAM)-based mycobacteria cell wall and its outer membrane components are TLR ligands, which selectively stimulate and generate classically activated M1 macrophages (Figure 2). *Mycobacterium* sp., although weakly Gram-positive (stained by the Ziehl–Neelsen method, acid-fast staining), differ from Gram-negative bacteria which contain LPS in their outer membrane, and they are exceptional, as their LAM-based membranes resemble LPS with respect to the induction of inflammatory responses. These inflammatory responses occur through recognition by the LPS-binding protein (LBP) and the soluble scavenger receptor CD14 (sCD14) which regulate cell toxicity. LAM-based mycobacteria species, such as MAP and bovine tuberculosis (caused by the *Mycobacterium bovis* bacillus Calmette–Guérin (BCG)), are mannose-capped LAMs (Man-LAMs), as has also been observed with the pathogen of Mbt, compared to the non-pathogenic forms of arabinosylated-LAM (Ara-LAM)-based mycobacteria [208,209].

### 7.2. Signaling Event Pathways and lncRNA Intervention

The molecular basis of antigen uptake by M cells has been progressively identified in the last decade in humans, although unresolved points are being investigated in ongoing research. The follicle-associated epithelium (FAE) covering mucosa-associated lymphoid tissue is important and is distinct from the villous epithelium in terms of its cellular composition and functions [207]. The excretion of interleukin-22 binding protein (IL-22BP) by dendritic cells at the subepithelial region results in the inhibition of the IL-22-mediated secretion of antimicrobial peptides by the FAE, while notch signaling from stromal cells underneath the FAE reduces goblet cell differentiation [207]. The latter events diminish the mucosal barrier functions, allowing luminal microorganisms to promptly gain access to the luminal surface of the FAE. Moreover, receptor activator of NFκB ligand (RANKL) from a certain stromal cell type induces differentiation into M cells that specialize in antigen uptake in the gut mucosa, where M cells initiate antigen-specific mucosal immune responses represented by the induction of secretory immunoglobulin A (S-IgA) [207]. When the inflammatory response progresses, IL-4, produced by type 2,T helper cells, stimulates M2 macrophage generation (Figure 2). In turn, M2 antagonizes the M1 effect to produce a net reduction in inflammation that is conducive to recovery from injury. In comparison with cattle, there are similarities in the regulation of homeostasis in M1/M2 bovine macrophages infected by MAP [19].

An important aspect of inflammatory response production is the participating receptors that internalize the pathogen. However, the intestinal microbiota is composed of a considerable population of microorganisms, which is maintained under balanced dynamic host–microbiota interactions, beyond which a dynamic equilibrium imbalance may occur, leading to dysbiosis of the enteric microbiota, as has been demonstrated in IBD and CD, but is not evident in UC patients [204,210]. In more detail, metagenomic research has indicated the presence of changes in bacterial species, while the phylum *Proteobacteria* was significantly increased and the phyla *Firmicutes* and *Bacteroidetes* were significantly reduced, representing a signature of the fecal microbiota in patients with CD [210]; more importantly, complications during mycobacterial infections have been proposed as a route for malignant pathogenesis. To expand the latter point further, and beyond the anticipated zoonotic risk that microbes and infectious agents present, they are also known causes of human cancers, while colorectal cancer (CRC) is a complication of UC and colonic CD, the two main forms of idiopathic inflammatory bowel disease (IIBD) [66]. The main receptors involved in MAP binding are (a) complement receptors and (b) integrin receptors. As shown in Figure 2, following phagocytosis and phagosome maturation, MAP bacteria interact with the host immune system and trigger cell signaling events to cause polarization into M1/M2 macrophages following infection by *Mycobacterium* sp. and subspecies such as MAP or Mbt. Phagocytic receptor binding allows MAP to be bound to CD14 on macrophages and dendritic cells, initiating signaling through TLRs such as TLR-2 and TLR-4 prior to antigen presentation, a common pathway for mammals. The antigen presentation and T helper cell activation produce either Th1 and/or a switch to Th2 cytokines, depending on the “immune challenges” and state of immunity of the host, but also include Th17 and T-Regs subsets [211]. Treg cells and Th17 have opposing activities but arise from a common precursor upon TGF-β stimulation that stimulates antigen presenting cells (APCs), macrophages, dendritic cells, fibroblasts, and endothelial cells to produce proinflammatory cytokines such as TNF-α, IL-1, IL-6, IL-8, IL-12, and IL-18 (Figure 2).

An important intervention in the protective cell signaling pathways is autophagy (macroautophagy), which is a cellular adaptive response and homeostatic mechanism to physiologic stressors such as starvation and inflammation that targets dysfunctional intracellular components for degradation via autophagic vesicles [212], having implications for immune-mediated gastrointestinal disorders such as CD. The formation of autophagic vesicles involves autophagy-related (ATG) gene products. ATG dysregulation has been implicated in immune disorders such as CD [213], and ATGs may serve as disease biomarkers [214]. Thus, autophagy function through *IRGM*, *ATG16L1*, and *ATG7* gene expression has been implicated in childhood-onset CD [215,216]. Autophagy and related genes can be explored further to understand their roles and mechanisms, and interest has been raised in terms of their roles in CD- and IBD-related pathologies in the discovery of additional disease biomarkers.

### 7.3. Coding Genes and LncRNAs Regulate the Pathological Phenotype Variables

The genetic and epigenetic variables influenced by environmental factors are the phenotype variables. Their regulated values are coordinated by the following factors: (a) genetic variations and polymorphisms, e.g., the genetic variability includes the associated susceptibility genes, such as *NOD2*, *SLC11A1*, and *ATG16L1* [217] (Gao et al., 2022), including the important autophagy-gene-associated functions, thus rendering the host with an increased susceptibility to infection and thereby a propensity to manifest the disease. IBD patients with the *ATG16L1* and *NOD2* mutations, as well as studies on murine models have demonstrated the disturbed secretory apparatus of P cells, resulting in defects of antibacterial autophagy [218]; (b) non-coding RNA regulators and their polymorphisms in combination with epigenetic factors (as discussed in Section 6 and Appendix A) alter gene regulation; and (c) specific ncRNA actions, such as the miR-31 binding potential on cytokine receptors, are crucial to inflammation control, as found in DSS-induced colitis [219].

In more detail, *NOD2* and the autophagy gene *ATG16L1* are expressed by an important type of cell, the Paneth (P) cells, in the intestinal mucosa upon bacterial challenge and inflammation [220]. P cell granule secretion into the lumen of crypts is governed by cholinergic and bacterial factors, such as the gene associated with IBD, *NOD2* [221], including the degranulation of P cells via the toll-like receptor 9 gene (*TLR9*). CD is characterized by a defective intestinal barrier towards intestinal microbes, while the cellular and molecular basis of this defect is likely to include P cells as major participants. P cells predominantly reside in the small intestine; however, P cells may also be induced by inflammation as metaplastic cells, owing to various stressors (e.g., cigarette smoking, stomach acid, excessive hormones, etc.) in other parts of the intestine, such as the colon. An important genetic contributor is the *NOD2/CARD15* gene, which is expressed apart from P cells in dendritic cells, macrophages, and intestinal epithelial cells (Figure 2); its risk variants are associated with lower levels of α-defensins in P cells, leading to impaired antimicrobial function. Importantly, autophagy gene *ATG16L1* risk variants compromise the autophagy function of P cells in patients with CD [222], which is the most well-studied target as a risk gene for CD in P cells so far [223]. P cells serve a dual function (1) by supporting the surrounding LGR5 positive stem cells via Wnt signals and (2) through their antibacterial secretion, while P cells seems to be the main source of IL17, also leading to TNF production, which is constitutively expressed in the P cells of the crypts [150]. Importantly, P cells have been used as a major therapeutic target in the intestine [223].

Important risk variants in murine models as well as IBD patients impair P cell function, leading to colitis; notably, the autophagy gene *ATG16L1* and *NOD2* mutations, with disturbed secretory P cells, result in antibacterial autophagy defects. In IBD, there is an abnormal goblet cell function, including the MUC2 and RELMβ proteins, while mucosal barrier dysfunction is represented by goblet (G) and P cell (P) functional disruption inside the intact intestinal epithelium (Figure 2). However, only 25% of IBD heritability has been explained by genetic studies, while 80–90% of GWAS-identified loci are confined to non-coding variations with pathogenic consequences via the modulation of gene expression [204]. Importantly, ncRNAs, which play a critical role in modulating host–microbe interactions, and, specifically, lncRNAs have been proposed as potential modulators of the host response to microbiome-linked pathologies such as cancers and obesity.

LncRNA regulators and genetic variations in combination with epigenetic factors play significant role in the gene regulation of proinflammatory genes. One of the roles of lncRNAs during infection is to regulate the polarization of M1 and M2 macrophages (Figure 2). Examples of lncRNAs produced in M1 macrophages are *HOTAIR*, *H19*, *MALAT1*, *lincRNA-COX2* (*PACER*), *ROCKI*, *GAS5*, and *CARLR* [224]. These lncRNAs can induce macrophage polarization into the M1 phenotype (primed by IFN-γ, TNF-α, LPS, and GM-CSF), while macrophage polarization into the M2 phenotype produces lncRNAs such as *PTPRE-AS1*, *KCNQ1OT1*, and *NEAT1* (primed by IL-4, -10, -13, TGF-β, miR-223) [225]. LncRNAs, such as *PACER* and *MALAT1*, directly interact with NF-κB to regulate target gene transcription, while others, such as *HOTAIR* and *CARLR*, indirectly regulate the NF-κB pathway by interacting with its upstream components. Given the effect of lncRNAs on NF-κB regulation, there is evidence of an association between lncRNAs (e.g., *CARLR*, *HOTAIR* etc.) and human diseases relevant to NF-κB dysfunction [224,225]. Bovine lncRNAs, such as *H19*, *DANCR*, *MEG3*, *MEG8* and *MIR221HG* (Figure 2), are differentially expressed upon MAP challenge and are upregulated, in contrast to *OIP5-AS1* (Table 3) (novel antisense transcript known as Cyrano), described earlier, which is associated with countless normal biological functions and numerous disease states [164,166], pleiotropic roles in normal homeostatic functions, and disease etiologies. Furthermore, reports on animals have shown that *OIP5-AS1* loss of function also disrupts embryonic development in zebrafish, which can be rescued by mammalian orthologs [226]. Thus, examining the genomic context and/or short regions of conservation in lncRNAs may be necessary for understanding their functions. Interestingly, as described in Table 3, MEG3 promotes bovine myoblast differentiation by sponging miR-135, while the upregulation of *MEG3* in ulcerative colitis can augment the protective effect of M2-macrophage-derived extracellular vesicles against UC while reducing inflammation [227,228,229]. In addition, the downregulation of MEG3 decreased M1 macrophage polarization and elevated M2 macrophage polarization by upregulating miR-223 in a mouse model of viral myocarditis (Figure 2) [230]. In cattle, the lncRNA *MIR221HG* (Table 3), which overlaps with miR-221 in the genome (owing to its name, which comes from the associated miR-221 followed by the initials of the host gene (HG)), and *ADNCR* are both novel lncRNAs that inhibit bovine adipocyte differentiation [231,232]. Thus, it could be suggested, following the similar regulatory potential as miR-221 in CD patients, that MIR221HG makes a comparable contribution to the progress of infection by MAP in JD.

## 8. The Prospect of LncRNAs in Contemporary Therapies of GI

IBD-related pathogenesis is a challenging subject for medicine to deal with in terms of the initial trigger of the disease, the diagnosis, the involved mechanism(s), and potentially, an effective therapy. Likewise, concerning JD, there is an ongoing and severe problem with animals at their productive age, whereby almost all animals develop critical pathological signs including weight loss, diarrhea, and secondary infections [233]. Reports are incessant concerning MAP infection in a wide range of hosts, including subjects not only from cattle but also from other important ruminants such as sheep, goat, deer, pigeons, etc., which are significant dairy product resources, or even non-ruminant species [233], which can easily transmit the MAP-resistant pathogen to other animals and the human population. A recent study explored the genetic relationship of *Mycobacterium avium* complex (MAC) isolates either between patients or between patients and their environment, confirming the probability of patient-to-patient transmission. They identified the hospital water distribution system as the main reservoir of non-tuberculous mycobacteria (NTM); this is especially important for nosocomial and healthcare-associated NTM infections and their transmission control [234]. Zoonosis is currently a biomedical and epidemiological risk with increasing interest and concern in society regarding health and safety with scientific, socioeconomic, and environmental impacts. Thus, despite the inconclusive evidence and lack of more specific studies and given the survival of MAP in the environment and water, is very likely that MAP is one potential MAC infectious agent that can be transmitted to humans through the healthcare-associated NTM bacteria load [234].

As mentioned earlier, another important parameter is the environmental basis of the disease, including pathogenic infection as one external trigger that precipitates intestinal inflammation [235,236], while recurrent pathogen infections in mice can develop as a consequential IBD-like syndrome [236]. Therefore, the first preventive strategy of the immune system is to minimize the risks of both inflammatory damage to the tissue and the development of chronic inflammatory diseases (e.g., CD), and the second is that pathogens need to be managed in ways that maintain the benefits of symbiosis in the intestinal flora in order to sustain the “harmonious” balance in the intestine [235].

There are a few examples presented below that support the rationale behind the idea of using lncRNA applications, particularly for disease diagnosis and therapy. Ge et al., illustrated that the level of lncRNA *ANRIL* (*CDKN2B-AS1*) can be used to distinguish patients with CD from healthy controls [237] and that *ANRIL* can serve as a biomarker under multiple conditions. In fact, changes in *ANRIL* expression are associated with the infliximab treatment response in patients with CD, as *ANRIL* expression in responders of infliximab treatment was increased, whereas that from non-responders remained unchanged. The latter effect demonstrates that *ANRIL* upregulation in the intestinal mucosa can act as a biomarker for assessing the response to infliximab treatment in patients with CD. Therefore, the latter effect can be considered evidence of lncRNA action and a predictor of the therapeutic response in IBD [237].

It is well-known that corticosteroids are commonly prescribed drugs for IBD, while glucocorticoids (GCs), in particular, show anti-inflammatory and immunosuppressive effects which are used to induce remission in UC patients, and also, they are of benefit in CD [238]. However, only ~20% of treated patients with GCs develop resistance to GCs, and 40–50% of patients maintain the clinical remission phase under medication with GCs, while a further 30% achieve partial remission. In poor- or non-responders to GCs, the levels of lncRNA growth arrest-specific 5 (*GAS5*) (Table 2 and Table 4) were found to be higher than those in good responders, showing evidence that GAS5 may be associated with GC resistance [239,240]. Furthermore, another study demonstrated that *GAS5* gene expression differs between GC-sensitive and GC-resistant cells, and *GAS5* is positively correlated with GC resistance in children with IBD [240,241]. Importantly, endogenous GAS5 affects the effectiveness of GCs, because it very likely accumulates in the cytoplasm by playing a role at the post-transcriptional level. Thus, based on the above evidence, *GAS5* may be considered a novel candidate biomarker with potential use in personalized GC therapy [240].

Currently, there are no approved lncRNA-targeting drugs; however, an increasing number of lncRNA-based approaches are in the preclinical phases, bringing the prospect of personalized medicine within reach in the future. LncRNAs can be targeted by adapting therapeutic approaches accordingly, and the following points may be taken into account, as illustrated below:Post-transcriptional RNA degradation pathways aimed at the knockdown of pathogenic RNAs could be an option, either by using siRNAs that elicit a DICER- and ARGONAUTE (AGO)-dependent cleavage pathway or by targeting the RNA of interest for degradation through an RNase H-dependent mechanism employing antisense oligonucleotides (ASOs).The modulation of lncRNA genes is also feasible by exploiting steric hindrances of the promoter under study using genome-editing techniques, e.g., CRISPR/Cas9 etc., andLoss-of-function, gain-of-function, and dominant-negative mutations have profoundly different effects on the protein structure, therefore providing great insight into the molecular mechanisms underlying genetic diseases. Thus, loss of function by steric inhibition of RNA–protein interactions, e.g., studying HuR binding or the disruption of its secondary structure using the RNA binding of small molecules or the aforementioned ASOs is another attractive approach.

A recent improvement in oligos technology is the use of modified ASOs to specifically bind to and block the lncRNA interaction interface, resulting in a loss of function. Also, small molecules that recognize unique RNA motifs and RNA structure-mediated interactions can serve as a useful tool for targeting lncRNAs.

The nuclear-localized *MALAT1* has been successfully downregulated in skeletal muscle through the systemic administration of ASOs and may have therapeutic benefits for cancer patients [242]. The subcutaneous administration of ASOs targeting *MALAT1* effectively inhibited human lung cancer cell proliferation in a mouse xenograft model as well as in A549 lung cancer cells, which showed a great reduction in the *MALAT1* expression level and decreased the migration ability in vitro [243].

As mentioned earlier (refer to Section 4.2), most lncRNAs (including lincRNAs) overlap with nearby ORFs or several lncRNAs can work as natural antisense transcripts (NATs) that overlap with protein-coding genes, affecting the transcription of their overlapping *cis-* genes [244]. As antisense lncRNAs (ASs) are particularly common and are present in up to 72% of the genomic loci, there is evidence of divergent transcriptions [245]; thus, ASs are transcribed from the introns of protein-coding genes, and NATs are transcribed from the complementary strands of protein-coding genes.

Thereby, an advantage that NATs hold for applications is the antisense targeting NATs, or “antagoNATs”, which provides a precise way of knockdown NATs to boost targeted endogenous gene expression. A recent study showed that the NAT *PDCD4*-antisense RNA1 (*PDCD4-AS1*), which is involved in breast cancer progression, positively regulates the expression and activity of the tumor suppressor *PDCD4* in mammary epithelial cells [246]. Elaborating further on the latter, in recent years, the lncRNAs that have emerged as potential therapeutic targets in breast cancer therapy are still promising. However, preclinical studies involving the ASO-based silencing of lncRNAs in breast cancer are very limited. Xing et al. showed that the therapeutic delivery of antisense locked nucleic acids (LNAs) specific to lncRNA Breast Cancer Anti-Estrogen Resistance 4 (*BCAR4*) (Accession No: NONHSAG018621.2 (NONCODE)) effectively suppresses metastasis in a breast cancer mouse model using locked nucleic acids (LNAs) targeting the lncRNA *BCAR4* and the knockdown of *BCAR4* by siRNA, inhibiting osteosarcoma tumorigenesis and lung metastasis in vivo [247], Moreover, Tian et al. recently showed the inhibition of the malignancy of esophageal squamous cell carcinoma cells in vitro and in vivo [248].

However, despite the new discoveries and methods of ncRNA detection and development in clinical diagnostics and therapeutic interventions, as mentioned in this section, measures are required to exclude any contamination in microbiome analyses and in therapeutic research. An important issue during in situ DNA and/or RNA isolation and detection from pathological tissue material, including during metagenomic analyses, especially in clinical practice, is to avoid the future expansion of the “contaminome” in order to ensure the fidelity of genomic information that is acquired in the laboratory, as recently reported [249], as a secure route to produce successful outcomes, as inaccuracy or contamination fosters a high level of risk in infectious diseases.

## 9. Conclusions

Recent advances in information technologies, in association with genetic and epigenetic discoveries, have revealed unknown folds in the regulatory networks of molecular pathogenesis for the intestine that promise to harness diseases of the gut and control therapeutic outcomes. A significant profile of molecules in the array of molecular networks, including the non-coding RNAs of long transcripts (>200 ntds), has recently been implicated in various diseases through significant regulatory roles.

In recent years, a limited number of proteins or peptides encoded by ncRNAs have been demonstrated to exhibit significant biological and pathological functions associated with the triggering and progression of intestinal barrier dysfunction and the importance of the intestinal microbiota. Numerous studies have shown a link between intestinal dysbiosis and IBD, as mentioned in Section 7.1. Although few studies have focused on ncRNAs in the modulation of dynamic host–microbiota interactions, the molecular regulators of ncRNAs in the intestinal microbiota are still not fully understood; likewise, the roles of intestinal microorganisms in initiating and facilitating the IBD-related pathologies are not known. Notably, many of the lncRNAs involved in JD and mycobacterioses are also major contributors to dysfunctional gene expression in humans and are mostly related to various types of cancer, including CRCs. A few examples are *MALAT1*, *CCAT1*, *CBR3-AS1*, and *LncRNA-MEG3*, whose functional details are shown in Table 2 and Table 3.

The effects of lncRNAs have become increasingly appreciated in research and their “discrete” contributions to cell homeostasis in physiological conditions and disease have started to unfold and unravel. Their profound effects on health and disease have promoted our scientific knowledge, which can be used for future research and for disease control. This research is of high priority and had apparent consequence during the scourge of the COVID-19 pandemic, which involved the SARS-CoV2 virus in 2020, as well as the influenza pandemic in 1918–1919; therefore, the best option is the development of preventive strategies to combat infectious diseases, as hesitation may put the human/animal population at risk of future illnesses.

## Figures and Tables

**Figure 1 ijms-24-13566-f001:**
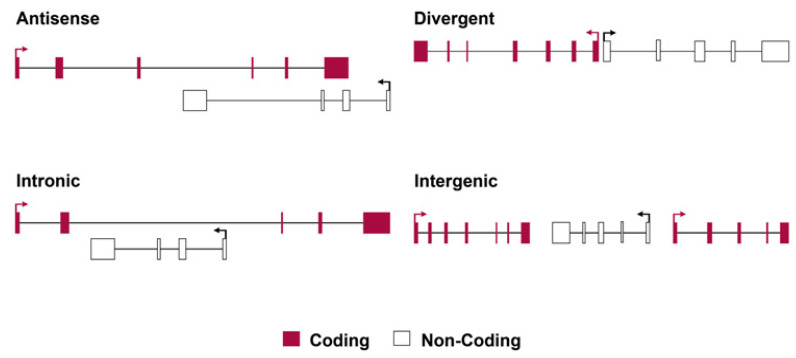
The organization of different types of lncRNAs in the genome depends on the location of the lncRNA sequence relative to nearby protein-coding genes and the direction of its transcription. Categories of lncRNAs are often defined by their locations, as shown above. **Antisense:** these are lncRNAs that initiate inside a protein-coding gene and transcribe in the opposite direction to overlapping coding exons. **Intronic:** these are the lncRNAs that initiate inside an intron of a protein-coding gene in either direction and terminate without overlapping exons. **Bidirectional:** these are lncRNA transcripts that initiate in a divergent fashion from a promoter of a protein-coding gene; the precise distance cut-off that constitutes bidirectionality is not defined but is generally within a few hundred base pairs. **Intergenic:** these are lncRNAs (also termed large intervening non-coding RNAs or lincRNAs) with separate transcriptional units from protein-coding genes.

**Figure 2 ijms-24-13566-f002:**
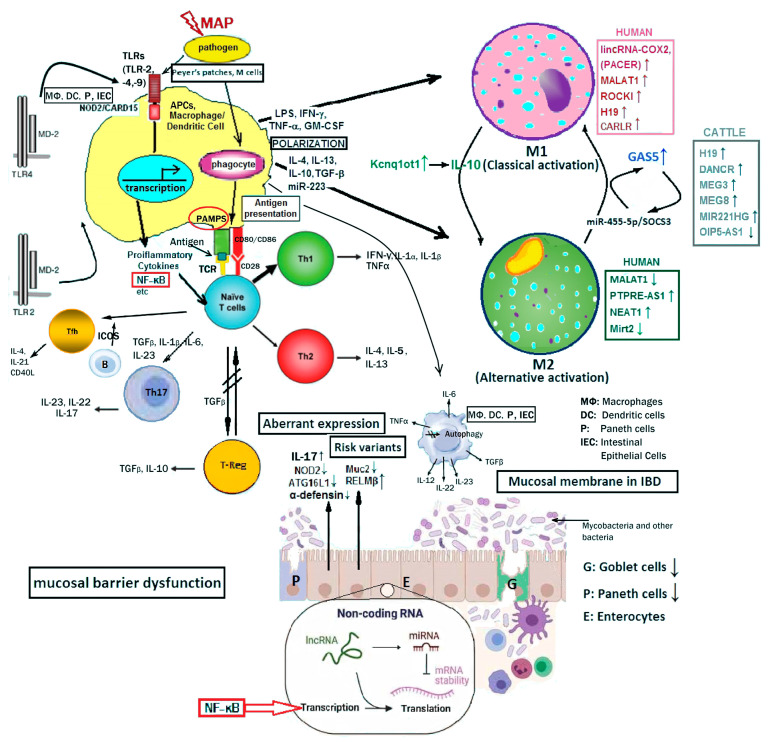
The MAP pathogen is the infectious trigger for the chronic development of JD and is also likely to be associated with IBD syndrome. Infection starts by MAP invasion and internalization in the M cells of Peyer’s patches within the intestinal mucosa, which play the role of antigen carriers. Mycobacteria (MAP) are bound to fibronectin and integrin receptors on M cells in the intestinal mucosa and to intraepithelial macrophages through complement receptors where they are phagocytosed. The later events produce “regulatory imbalance” in non-coding RNAs (ncRNAs) which play a critical role in modulating host–microbe interactions; thus, lncRNAs have been specifically proposed as potential modulators of the host response to microbiome-linked pathologies, such as cancers and obesity. Thus, one of the roles of lncRNAs during infection is to regulate the polarization of M1 and M2 macrophages. The corresponding lncRNAs that are produced to polarize either M1 or M2 macrophages in humans are shown inside the colored boxes (pink and green, respectively). (Detailed descriptions of the picture and terms are found inside the text). MΦ, macrophages, DC, dendritic cells; P, Paneth cells; IEC, intestinal epithelial cells. Arrows next to the genes or cells types: upwards or downwards arrows show increases or decreases in the number of cells or gene expression levels, respectively. All the other arrows show direction or end product.

**Table 1 ijms-24-13566-t001:** LncRNA genes and transcripts of five species retrieved from the NONCODE database of animals and the GENECODE and other databases (see text).

NONCODE	Human	Mouse	Cow	Pig	Chicken
**Genes**	96,411	87,890	22,227	17,811	9527
**Non-Coding Transcripts**	173,112	131,974	23,515	29,585	12,850
**GENECODE, NCBI, and Ensembl**	**Human**	**Mouse**	**Cow**	**Pig**	**Chicken**
**Coding Genes**	19,988	21,833	21,880	21,303	16,878
**Coding Transcripts**	87,814	59,138	43,984	63,041	39,288

Note: The distribution of the protein coding and non-coding information of each genetic element is in accordance with GENCODE and NONCODE statistics and genomic information from annotation releases 110 and 106 for humans and cattle, respectively, which were retrieved from the NCBI. Numbers are presented in thousands with commas as separators.

**Table 2 ijms-24-13566-t002:** Characterized and novel lncRNAs involved in Crohn’s Disease (CD) and Ulcerative colitis (UC) published or in silico searched and cited.

Disease	Source	Gene Expression Change	Transcript/Gene Name	Gene ID	Cited by, Reference	Location (hg38)	Reported Mechanism
DSS-induced colitis	Mice serum and tissues	Upregulated	NEAT1	NEAT1	[122]	chr11	Modulated intestinal epithelial barrier
UC & CD	Colonic tissues	Upregulated	**CCAT1**	**lnc-APPL2-1,** **lnc-BMP6-106,** **lnc-FAM84B-15**	[123]	chr12, chr6, chr8	Increased barrier permeability
UC	Colonic tissues	Upregulated	ANRIL	CDKN2B-AS1	[124]	chr9	Promoted inflammatory cytokines and chemokines production
UC	Colonic tissues	Downregulated	CDKN2B-AS1	CDKN2B-AS1	[125]	chr9	Enhanced the barrier formation
UC	Colonic tissues	Upregulated	IFNG-AS1	IFNG-AS1	[126]	chr12	Enhanced inflammation
UC	Colonic tissues	Upregulated	H19	H19	[127]	chr11	Disrupted intestinal epithelial barrier function
CD	Plasma	Upregulated	**ENST00000466668**	**GUSBP2**	[128]	chr6	
			**ENST00000422548**	**AL022100.2**		chr1	
			**ENST00000502712**	**RP11-68L1.2**		chr3	
			**ENST00000425364**	**lnc-REV3L-1**		chr6	
				**LINC01272**			
			NR_037605	GAS5-AS1		chr1	
			**ENST00000562996**	**FIGNL2** **-DT divergent transcript**		chr12	
			**NR_038927**	**DDX11-AS1**		chr12	
			TCONS_00014043(*)	-		chr7	
			TCONS_00012771(*)	-		chr6	
			**ENST00000569039** **(novel transcript)**	**Antisense to C16orf53**		chr16	
		Downregulated	uc001ody.3	-		chr11	
			**ENST00000575787**	**ALOX12P2**		chr17	
			ENSGO0000259472	**LOC80154** **(uc010bmo.1** **)**		chr15	
			**ENST00000509252**	**TRIM52-AS1**		chr5	
			ENST00000413954	**AC064871.3**		chr2	
			**ENST00000431104**	**RNF217**		chr6	
	Ileal Biopsy Terminal Ileum		**ENST00000452481.1**	**HNF4A-AS1**	[129]	chr20	
			**ENSGO0000245060.2** **(uc011dhd.3)**	**LINC00847**		chr5	
			**TCONS_00020749** **(*)**	-		chr12	
			**NR_027074**	**LINC00928**		chr15	
			TCONS_00027621(*)	-		chr19	
CD		Upregulated	**ENST00000460164.1**	**lnc-BRF1-9**		chr14	
			ENST00000532855.1	MMP12		chr11	
			ENST00000326227.5	MMP12		chr11	
	Whole Blood		**ENST00000419897.1** **ENST00000535913.2**	**SLC12A5-AS1**		chr20	
			**ENST00000520185.1**	**ENST00000520185**		chr11	
			**ENST00000526690.1**	**lnc-ZNF705D-2**		chr8	
			**ENST00000445003.1**	**lnc-CEBPB-13**		chr20	
			**ENST00000522970.1**	**lnc-ADAM2-1**		chr8	
			**ENST00000524555.1**	**lnc-SAA2-SAA4-1**		chr11	
			**ENST00000429315.2**	**KIF9-AS1**		chr3	
		Downregulated	**ENST00000432658.1**	**DPP10-AS1**		chr2	
			**ENST00000401008.2**	**lnc-NBPF11-2**		chr1	
			**ENST00000553575.1**	**DIO3OS**		chr14	
			**ENST00000554694.1**	**DIO3OS**		chr14	
			**ENST00000557532.1**	**DIO3OS**		chr14	
			**ENST00000557109.1**	**DIO3OS**		chr14	
			ENST00000422420.1	CDKN2B-AS1		chr9	
			ENST00000428597.1	CDKN2B-AS1		chr9	
			**ENST00000554441.1**	**DIO3OS**		chr14	
			**ENST00000554735.1**	**DIO3OS**		chr14	
CD	Ileal tissues	Upregulated	ENST00000487539.1_1	MMP12	[130]	chr11	Involved in the pathogenesis of CD
			**ENST00000409569.2_1**	**MIR4435-1HG**		chr2	
			**ENST00000392442.6_1**	**RSRC2**		chr12	
		Downregulated	**ENST00000524613.5_1**	**ENSG00000254645**		chr11	
			**ENST00000465605.5_1**	**ENSG00000234539**		chr6	
CD	Blood	Upregulated	DQ786243	-	[131]	chr1	Affected CREB and Foxp3 expression and regulated Tregs function
UC	Colonic tissues	Upregulated	**ENST00000460164.1**	**lnc-BRF1-9**	[132]	chr14	
			ENST00000532855.1	MMP12		chr11	
			ENST00000326227.5	MMP12		chr11	
			**ENST00000419897.1** **ENST00000535913.2**	**SLC12A5-AS1**		chr20	An IBD that has material basis in variation in the chromosome region 12q15.
			**ENST00000429315.2**	**KIF9-AS1**		chr3	
			**ENST00000526690.1**	**lnc-ZNF705D-2**		chr8	
			**ENST00000524555.1**	**lnc-SAA2-SAA4-1**		chr11	
			**ENST00000476886.1**	**CLRN1-AS1**		chr3	
			**ENST00000517774.1**	**TNFRSF10A-DT**		chr8	
			**ENST00000578280.1**	**lnc-IGFBP4-1**		chr17	
		Downregulated	ENST00000422420.1	CDKN2B-AS1		chr9	
			ENST00000428597.1	CDKN2B-AS1		chr9	
			ENST00000585267.1	CDKN2B-AS1		chr9	
			ENST00000580576.1	CDKN2B-AS1		chr9	
			ENST00000577551.1	CDKN2B-AS1		chr9	
			ENST00000581051.1	CDKN2B-AS1		chr9	
			ENST00000582072.1	CDKN2B-AS1		chr9	
			**ENST00000401008.2**	**lnc-NBPF11-2**		chr1	
			**ENST00000432658.1**	**DPP10-AS1**		chr2	
			ENST00000421632.1	CDKN2B-AS1		chr2	
		Upregulated	ENSG00000070190	**D** **APP1** **-001**		chr4	Gene hypomethylated. May act as a B-cell-associated adapter that regulates B-cell antigen receptor (BCR)-signaling downstream of PI3K
UC	Colonic tissues	Upregulated	**BC012900**	-	[133]	chr8	Regulated intestinal epithelial cells apoptosis
			**AK001903**	-		chr7	
			**AK023330**	-		chr9	
		Downregulated	**BC029135**	-		chr10	
			CDKN2B-AS1	CDKN2B-AS1		chr9	
UC	Colonic tissues	Downregulated	ENST00000647780.1	MEG3	[134]	Chr14	LncRNA MEG3 could improve ulcerative colitis by upregulating miR-98-5p-Sponged IL-10
UC				**BC062296-**		chr21	
UC	colonic epithelial cell	Upregulated	ENST00000619449.2	MALAT1	[135]	chr11	promotes ulcerative colitis by upregulating lncRNA ANRIL
UC	Colonic tissues	Upregulated	H19	H19	[136]	chr11	Promoted mucosal regeneration
UC	MSICT	Upregulated	H19	H19	[137]	chr11	Regulated intestinal epithelial barrier by interacting with HuR
DSS-induced colitis, IBD	Mice colonic tissues	Upregulated	**CRNDE**	**lnc-IRX3-80**	[138]	chr16	Promoted epithelial cells apoptosis
UC	Colonic tissues	Downregulated	**SPRY4-IT1** **(NR_131221.1)**	**SPRY4-IT** **1**	[139]	chr5	Regulated intestinal epithelial barrier function
UC	Mice small intestinal tissues	Upregulated	**uc.173**	**uc.173**	[140]	-	Stimulated intestinal epithelium renewal & regulation intestinal epithelial barrier function
DSS-induced injury	IEBM	Upregulated	**PlncRNA1**	**PlncRNA1**	[141]	chr21	Regulated tight junction proteins
UC	Colonic tissues	Upregulated	IFNG-AS1	IFNG-AS1	[142]	chr12	Regulated pro-inflammatory cascade
UC	Blood and monocytes	Upregulated	**MROCKI** **(uc003pwa.3)**	**MROCKI** **(ENSG00000227502.2)**	[142]	chr6	Promoted inflammatory cytokines and chemokines production
UC	Mice colonic tissues	Upregulated	**HIF1A-AS2**	**HIF1A-AS2, lnc-TMEM30B-9**	[143]	chr14	Negatively regulated intestinal inflammation and exerts oncogenic functions in colorectal cancer
EC, IBD	Colonic tissues	Downregulated	**NRON**	**NCRNA00194**	[144]	chr9	NRON is a repressor of NFAT component LRRK2 in EC.
SLE, Lupus nephritis	Innate & Acquired immune system, Blood, Nervous system	Upregulated	**ENST00000607592.2**	lincRNA-COX2	[145]	chr3	Control of inflammatory response. Induction followed by TLR2 and TLR4 stimulation in NFκB dependent manner. Maintain homeostasis within the lung
UC & TNF-α-treated HT-29 cells	Colonic tissues	Downregulated	**ENST00000644773.3**	**TUG1**	[146]	chr22	TUG1 played protective role in UC by preventing TNF-α-induced cell injury and inflammation
UC, IBD	PBMCs	Downregulated	**LINC01882**	**lnc-PTPN2-2**	[147]	chr18	Involved in T cells activation and IL-2 expression
DSS-induced colitis	Mice colonic tissues	Downregulated	NEAT1	NEAT1	[148]	chr11	Regulated by 5-ALA and involved in PDT therapy treated colitis
Chemicaly induced colitis, UC, inflammed lung	Colonic tissues	Upregulated	**ENST00000306042.9**	**PTPRE-AS1**	[149]	chr10	controls macrophage function, exacerbates chemical-induced colitis
Celiac, intestinal disease	Colonic tissues	Upregulated	**ENST00000518376**	**LncRNA-CARL,** **LINC00990**	[150]	chr8	In celiac disease patients, increased levels of the Carlr transcript were detected in the cytoplasm, alongside elevated expression of NF-κB pathway genes

Note: *lncRNA* long noncoding RNA, *IBD* inflammatory bowel disease, *CD* Crohn’s Disease, *UC* Ulcerative Colitis, *DSS* dextran sulfate sodium, *qPCR* quantitative real-time PCR, *NEAT1* nuclear paraspeckle assembly transcript 1, *5-ALA* 5-aminolevulinic acid, *PDT* photodynamic therapy, *RNAseq* RNA sequencing, *CRNDE* colorectal neoplasia. differentially expressed, *CCAT1* colon cancer–associated transcript–1, *CREB* cAMP response element binding protein, *Foxp3* Forkhead box P3, *IL-2* Interleukin-2. (*) lincRNA and transcripts of uncertain coding potential (TUCP). *IBD* Inflammatory Bowel Disease, *EC* Experimental Colitis, Novel or uncharacterized/unchategorized transcripts or Ensemble processed transcript, **in bold**. (Table constructed from Lin et al., 2020, Ghafouri-Fard et al., 2020 [151,152] and in silico search from several databases, as described in the text).

**Table 3 ijms-24-13566-t003:** Newly characterized and novel bovine lncRNA information associated with bovine JD and/or mycobacterial disease retrieved from related platforms and databases (see text). Details include the interaction type, targets, NCBI accession numbers, and descriptions supported by citation IDs (PMIDs).

ID	Name	Chromosome	Disease Name	Interaction	Interaction Target	NCBI Accession	Description (Interaction)	Description (Function)	PMID
EL0129 (--)	ADNCR	13	N/A	binding	miR-204	NR_137293	LncRNA ADNCR functions by targeting miR-204 to significantly regulate the expression of the target SIRT1 gene in preadipocytes at both the mRNA and protein levels, thereby inhibiting adipogenesis.	Identification of a novel polymorphism in the bovine lncRNA ADNCR gene and its association with growth traits. LncRNA ADNCR suppresses adipogenic differentiation by targeting miR-204.	29631473 27156885
ENSG00000226950	DANCR	4	N/A		miR-4449 (ENSBTAT00000064646)	NR_131910.1		miR-4449 is enriched in the serum exosomes of diabetic kidney disease patients. These exosomes regulate the expression of proinflammatory cytokines, ROS levels, and pyroptosis through miR-4449	17488528
EL1185 (--)	H19	29	bovine mastitis	N/A	N/A	NR_003958	All observations imply that lncRNA H19 modulates the TGF-β1-induced epithelial to mesenchymal transition in BEC through the PI3K/AKT signaling pathway, suggesting that MEC might be one source of myofibroblasts in vivo in the mammary glands under inflammatory conditions, thereby contributing to mammary gland fibrosis.	The overexpression of lncRNA H19 changes the basic characteristics and affects the immune response of bovine mammary epithelial cells.	29062612
EL1909 (--)	lnc133b	N/A	N/A	binding	miR-133b	N/A	IGF1R is an important target gene of miR-133b in bovine skeletal muscle satellite cells. lnc133b increases IGF1R expression by the “sponge” miR-133b.	Lnc133b promotes bovine skeletal muscle satellite cell proliferation and represses differentiation by acting as a ceRNA for miR-133b.	28757453
EL1922 (--)	lnc403	N/A	N/A	regulation	KRAS, Myf6	N/A	lnc403 negatively regulates the expression of the adjacent gene Myf6 and positively regulates the expression of the interaction protein KRAS.	A novel lncRNA, lnc403, involved in bovine skeletal muscle myogenesis by mediating KRAS/Myf6.	32387386
EL2003 (--)	lncKBTBD10	N/A	N/A	regulation	KBTBD10	N/A	LncKBTBD10 could induce a decrease in the KBTBD10 protein and further affect bovine skeletal muscle myo genesis.	A novel long non-coding RNA, lncKBTBD10, involved in bovine skeletal muscle myogenesis.	30465303
EL2170 (--)	lncRNA-TUB	N/A	bovine mastitis	N/A	N/A	N/A	A novel long non-coding RNA regulates the immune response in MAC-T cells and contributes to bovine mastitis.		30771271
EL2345 (--)	LRRC75A-AS1	N/A	bovine mastitis	regulation	LRRC75A	N/A	LRRC75A antisense lncRNA1 knockout attenuates inflammatory responses of bovine mammary epithelial cells.	LRRC75A-AS1 may protect LRRC75A from degradation by binding to its CDS region.	31929753
EL2406 (--)	MDNCR	N/A	N/A	binding	miR-133a	N/A	MDNCR was observed to directly bind to miR-133a with 32 potential binding sites	MDNCR binding to miR-133a promotes cell differentiation by targeting GosB in cattle primary myoblasts.	30195797
EL2412 (ENSG00000214548)	MEG3	21	N/A	binding	miR-135B	NR_037684	LncRNA-MEG3 promotes bovine myoblast differentiation by sponging miR-135.	LncRNA-MEG3 was verified as a miR-135 sponge. Overexpression of miR-135 markedly inhibits wild-type lncRNA-MEG3. Also, it has prognostic value in cervical cancer in humans.	30887511 28057015
EL2415 (ENSG00000258399, ENSG00000225746)	MEG8	21	N/A	N/A	N/A	NR_146189		Three SNP sites were identified in these three lncRNAs. They showed monoallelic expression in the analyzed tissues, suggesting that they may be imprinted in cattle.	27925264
EL2417 (--)	MEG8-IT3	N/A	N/A	N/A	N/A	N/A		Similar SNP effect to the previous variant detailed above	27925264
EL2451 (--)	MIR221HG	N/A	N/A	N/A	N/A	N/A		MIR221HG Is a novel LncRNA that inhibits bovine adipocyte differentiation.	31887993
EL2668 (--)	NONBTAT017009.2	N/A	N/A	N/A	N/A	N/A		MiR-21-3p centric regulatory network in dairy cow MEC proliferation.	31532202
EL3546 (--)	TCONS_00041733	N/A	N/A	co-expression	EFNA1	N/A	We found that TCONS_00041733 lncRNA targets the node gene EFNA1 (ephrin A1), which is involved in male reproductive physiology.	Integrated analysis of mRNAs and lncRNAs in the semen from Holstein bulls with high and low sperm motility.	30765858
ENSBTAG00000010766	OIP5	10	N/A, cancer (human)	expression and binding	ENST00000507296.1 lncRNA, MIS18 binding protein 1, HuR, mir-424		The RNA-binding protein HuR, which enhances cell prolifer- ation, associates with OIP5-AS1 and stabilizes it. mir-424 interacts with OIP5-AS1 and competes with HuR for binding to OIP5-AS1.	OIP5-AS1 reduces cell proliferation and serves as a sponge or a competing endogenous (ce)RNA for HuR, restricting its availability to HuR target mRNAs and thereby repressing HuR-elicited proliferative phenotypes.	26819413
EL3885 (ENSG00000229807)	XIST	X	glioma	binding	miR-34a, miR- 429	XR_001495596, XR_001495595, XR_001495594	XIST serves as a ceRNA for miR-34a through sponging miR-34a, competing with MET for miR-34a binding, and finally, modulating thyroid cancer cell proliferation and tumor growth. The MiR-429 inhibitor restores the metastatic and proangiogenic abilities of gliomas abolished by silencing XIST.	LncRNA XIST promotes glioma tumorigenicity and angiogenesis by acting as a molecular sponge of miR-429. The XIST/miR-34a axis modulates cell proliferation and tumor growth of TC. XIST expression is promoted by the activated NF-κB pathway; in turn, XIST generates (-) feedback to regulate the NF-κB/NLRP3 inflammasome pathway.	30463570 29187887 30463570 30362186
EL3948 (--)	XLOC_059976	N/A	N/A	N/A	N/A	N/A		XLOC_059976 could be used as candidate marker for milk protein content prediction.	30105049

Note: (ENSG): Ensembl accession number of the gene. (--): The Ensembl accession number is not available, only the EL number is available (first column). EL number: Accession is shown according to the Database Resources of the National Genomics Data Center, China National Center for Bioinformation in 2022, EVLncRNAs2.0. Data from NONCODE, EVLncRNAs2.0, LNCipedia, LncRNADisease v2.0 (see text). JD: Johne’s disease; BEC: bovine epithelial cells; MEC: mammary epithelial cells; SNP: single nucleotide polymorphism; TC: thyroid cancer.

**Table 4 ijms-24-13566-t004:** A general list of LncRNAs and their specific features is shown. These LncRNAs are involved in human IBD-pathologies, CD, UC, and MAP-infected cattle (JD).

Species	Human and Cattle			Human				Cattle
Pathology	Immune Regulation/ Deregulation	IBD-Related Pathologies	Additional lncRNAs in the CD (Dataset GSE75459) (§)	Crohn’s Disease/Upregulated	Crohn’s Disease/Downregulated	Ulcerative Colitis/Upregulated	Ulcerative Colitis/Downregulated	Johne’s Disease
	** *H19* **	CBR3-AS1	**Upregulated**	*GAS5-AS1*	DPP10-AS1	** *H19* **	NRON	** *H19* **
	DQ786243	GUSBP2	GUSBP3, 14, 15, 16	DQ786243	ALOX12P2	ANRIL (†)	DPP10-AS1	ADNCR
	BC012900	*GAS5-AS1*	FIGNL2-DT	GUSBP2	*DIO3OS*	** *KIF9-AS1* **	lnc-PTPN2-2	** *DANCR* **
	*CDKN2B-AS1*	NRON	GAS5-AS1	TALAM1	*CDKN2B-AS1*	** *IFNG-AS1* **	** *CDKN2B-AS1* **	lnc133b
	** *NEAT1* **	BC012900	**Downregulated**	DDX11-AS1	TRIM52-AS1	*NEAT*1 **(*)**	***NEAT*1 (*)**	lnc403
**LncRNA**	*CCAT1*	*LINC01272*	ALOX12P2 var1, 2	*CCAT1*	*HNF4A-AS1* (#)	*CCAT1*	RNF217	LRRC75A-AS1
	** *MALAT1* **	*DIO3OS*	ALOX12E	** *MALAT1* **	CDKN2B-AS1	** *MALAT1* **	SLC12A5-AS1	** *MEG3* **
	lincRNA-Cox2	DDX11-AS1	GOLGA2P8	lnc-MICAL2-1		*FOXD2-AS1*	** *SPRY4-AS1* **	MEG8
	** *DANCR* **	*IFNG-AS1*	** *MALAT1 var1,2,3* **	SLC12A5-AS1		SLC12A5-AS1	Mirt2	** *OIP5-AS1* **
	**MEG3**	*KIF9-AS1*	TRIM52 var1,2,3	*LINC01272* (#)		CRNDE (DSS)	** *MEG3* **	MDNCR
	**OIP5-AS1**	CRNDE (DSS)	TALAM1	lncRNA-CARL		*MROCKI*	** *TUG1* **	lncKBTBD10
	uc.173					uc.173		** *MALAT1* **
	lncRNA-CARL					lncRNA-CARL		

Note: (*) NEAT1 presents alternate behaviors, as both up- and downregulated (see text), (§): LncRNAs have been identified by Microarray GEO datasets (GSE75459, NCBI), (†) ANRIL (isoform of CDKN2B-AS1), CD: Crohn’s disease, UC: ulcerative colitis, JD: Johne’s disease, DSS: dextran sulfate sodium-induced colitis in a mouse model, MAP: Mycobacterium avium ssp. paratuberculosis. **Bold fonts**: shared regulatory lncRNAs for both species. (#) Significant correlations between *LINC01272* and *HNF4A-AS1* expression were found (see text). *Italic fonts:* Highly cited and potential biomarkers (see text).

## Data Availability

Further data in this study are available upon request from the authors.

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
