# Peer review of "Long Non-Coding RNAs and Their “Discrete” Contribution to IBD and Johne’s Disease—What Stands out in the Current Picture? A Comprehensive Review"

_ijms, 2023, doi:10.3390/ijms241713566_

Round 1
Reviewer 1 Report (Previous Reviewer 3)
Comments and Suggestions for Authors
The author improved the manuscript, but it still results too long and probably redundant, and the conclusions are too much hypothetic in my point of view.
Principal concerns remain:
I think it should be reduced significantly, focusing more on the main topic, while everything that concerns long noncoding RNAs in a particular way should be sent elsewhere.
In addition, the role of diet in expression should be considered more specifically, as this could give false results if they are used as biomarkers.
Comments on the Quality of English Languageit needs revision
Author Response
Cover Letter
Response_to_Reviewers_Triantaphyllopoulos_ijms-2549260
Dear Reviewer
Thank you very much for the constructive comments from the reviewers of this comprehensive review re-entitled, according to the suggestion of the Reviewers: “Long Non-coding RNAs and Their “Discrete” Contribution in IBD and Johne’s Disease. What stands Out in the Current Picture? A Comprehensive Review”.
In the revised manuscript, all the comments answered, the necessary information and suggested changes are provided to address each point made by the reviewers. I believe the new information and changes of the manuscript have significantly strengthened our manuscript and we thank the editors and reviewers for their helpful suggestions. Below, you will find, point-by-point, the answers and recommended changes suggested from the three reviewers. The reviewer’s comments are shown in blue fonts and my response in black fonts.
Also I would like to inform the reviewer and editors that the line numbering system in the template that was attached in the journal and email doesn’t appear in my word (possibly due to old Word version) and only in the pdf file, which I used as a guide for this revision. Therefore I sent my revised manuscript in plain word format.
The files that I have included in the Zipped folder named
Triantaphyllopoulos_KA(ijms-2549260_Revised) were prepared for each Reviewer and contain the following files and folders:
Manuscript: Triantaphyllopoulos_K.A. (highlighted changes)
Manuscript: Triantaphyllopoulos_K.A. (accepted changes)
Response_to_Reviewers_Triantaphyllopoulos(Reviewer_1)
Folder: LncRNA TABLES(Revised) which contains the following:
Table 1(tracked changes)
Table 1(accepted changes)
Table_2(tracked changes)
Table_2(accepted changes)
Table 3(tracked changes)
Table 3(accepted changes)
Table 4(Rebuilt)
LncRNA FIGS(Revised) which contains the following:
Figure 2
Following due consideration of my revised manuscript, I trust that you will find it suitable for publication in IJMS.
Thank you
Yours sincerely
The author
Kostas A. Triantaphyllopoulos
Reviewer
Comments and Suggestions for Authors
The author improved the manuscript, but it still results too long and probably redundant, and the conclusions are too much hypothetic in my point of view.
Principal concerns remain:
I think it should be reduced significantly, focusing more on the main topic, while everything that concerns long noncoding RNAs in a particular way should be sent elsewhere.
- I thank the reviewer for this suggestion. I have attempted to minimise, organise and put in place all the requirements in the resubmission, omitting several areas and the bulk of the body text, so the manuscript was reduced substantially in size. This has caused an arduous work to put back the remaining pieces. Therefore, the manuscript since last time has been shortened significantly and revised to focusing on the major points of interest and areas in present and future research and as much as possible informative as the first version.
It was a hard work to rearrange Figures, Tables and Supplementary Material, as it is now, and to keep the information up to a high standard, trying not removing the most important and updated information in the field, which is expanding rapidly and has to be shown.
To my humble opinion, more severe cut for this manuscript will eliminate any new piece of information pertinent to these pathologies in the context of the lncRNA intervention in these diseases, which was the core value of this manuscript and the aim was to disseminate this information to the scientific community.
In more detail, in this version, only 4 Tables and 2 Figures are presented, (compared to 10 and 9 in the old version), respectively (one of the current figures is new, which is Figure 2).
Graphical abstract remains the same. The length of the manuscript has been substantially shortened in 26 pages, 14065 words and 249 references compared to the previous version of 70 pages of 38817 words and 522 references. (Pages exclude References and Abstract). More severe cut of this manuscript will damage the flow of the sections and any new piece of information pertinent to these pathologies, the novel lncRNA lists, dataset and gene network analysis of studies that added new findings and value in the manuscript, and these were the aims of this work.
Also, Tables 2, 3 and 4 as well as Figure 2, were revised substantially.
In the Supplementary Material, 2 Tables remain (compared to 8 previously), and 2 Figures (compared to 4 Figures previously), and these are the gene network graphics from the LncACTdb 3.0 database analysis, considering these as important and indispensible for the reader. In this revision the Supplementary Material remained the same as none of the Reviewers had comments, although they were checked for errors. Thank you.
- I also thank the reviewer for the second half of the first suggestion by the reviewer, i.e. dividing the manuscript in two independent articles. I strongly believe, and the “birth” of this idea was to create a review including new evidence from various sources and personal analyses, concerning the involvement of lncRNAs in gene regulation of these pathological phenotypes. The example was found in the area of gastrointestinal inflammatory diseases including IBD and JD, which I was immensely interested in and involved in lab research several years ago as a much as I could balancing my limited time with my teaching duties.
Thus, this comprehensive review was created and concerning to long non-coding RNA alterations and regulation by associated factors found so far, in these disease subtypes which shape the current molecular profile in these complex pathologies.
I also intended to discuss this challenging subject under the new findings of noncoding RNA regulatory capacity. There is supportive evidence in favour or against zoonosis which is crucial to be complemented by the information of the updates, including the lncRNAs that support the trigger by bacterial infection in IBD or according to others’ opinion the opposite, in order, under compelling evidence, to divert the opinions and diagnosis of this unresolved disease from the infectious agent, should the expression patterns of these potential biomarkers, may not be specifically expressed. The latter elucidation point would be the landmark and the desired prospect prior to the hard work which is required and will lead to a diagnostic tool discovery or molecular targeting for therapy in the context of lncRNAs, and that is the message that we attempted to put across presenting this information.
Therefore, this review requires to include both concepts including a short introduction/background for the diseases in concern but always under the umbrella of long non-coding RNA regulation and mediation with its update that will be also followed in the future by the researcher to monitor the progress in the field with more updates. I hope the reviewer like better this short newly revised version of the manuscript may kindly share the view of the author. Thank you.
Therefore, the main topic and title of this comprehensive review is principally lncRNA action and the disease focus of lncRNA action, is on 2 certain pathologies, i.e. IBD and JD, but still these pathologies are closely related to the lncRNA action, which is the main subject of this study. Thus, the scope of this review cannot be purely divided to another standalone pathobiology article. Notably, more than 70 % of the manuscript, including introduction and other sections, database searches, including analyses, discoveries, role of lncRNAs, mechanisms, expression and regulation of the established and novel lncRNAs with their contribution in these pathologies.
In order to divide this work in pathologies in two, has to be a new project writing a new article from scratch and dealing purely with these diseases that will entail several microbiological, epidemiological, pathological and clinical aspects. I believe, this would be a new piece of work with completely different frame than the current work, which has to be built and enriched with the relevant information. The current review is deprived from this information which belongs to different discipline and does not contain such or is very limited and not included in the current manuscript, because the aim for this study was different. Thereby, this review does not include adequate material, clinical or pathophysiology description or opinions of medical and veterinary cases of IBD and/or JD, to support even the precursor of another pathology review or “main topic” review -as named by the reviewer- related to IBD/JD; and if proceeding to this direction will damage the initial idea of the review and will delay the suggested 2 separate submissions quite far from now, currently they are far from completion.
In addition, the role of diet in expression should be considered more specifically, as this could give false results if they are used as biomarkers.
- I thank the reviewer for this suggestion. I agree with the reviewer for the role concerning diet in gene expression, although this manuscript has not been implicated with diet biomarkers etc and made sure not to describe the variable of the diet in these gastrointestinal pathologies.
Therefore, in the current version the diet as a variable in gene expression is not mentioned but only in a phrase that contains Life Epigenetics in a general context, and not included potential biomarkers affected by diet or not examining dietary habits or calories uptake concerning gene expression.
I totally agree with the Reviewer from my experience in experiments that have performed in the animal department, which are in agreement with the reviewers’ opinion and I believe, that dietary metabolic genes we studied (which are not quoted here in this review), under conditions such as calorie-restriction etc, have not shown the expected phenotypic results as with protein-coding genes (e.g. for meat quality traits), while other indicators (e.g. histone deacetylases, noncoding genes or epigenetic complexes) show significant modulation changes that could have the potential to be biomarkers.
Also, in this review, which is also emphasized the point of the early stages current research is regarding biomarkers, that biomarker specificity is a difficult topic to conclude. More studies are required to establishing new lncRNAs biomarkers, and inflammatory diseases are a difficult parameter as perturbation of variables such as metabolic or diet factor changes often towards unpredictable gene expression.
The structure of the manuscript was also changed in several sections, as they have been merged and summarized and many parts from the previous version shortened and moved in other places minimize the length of the manuscript. Section 7 is new and it was important, as it was missing from the previous version unifying the principle of infection that triggers acute and chronic inflammatory mechanisms causing imbalance and destabilization of the regulatory work of lncRNAs that also infectious triggers do, which are described in an holistic point of view.
Thank you.
Comments on the Quality of English Language
it needs revision
I have checked the overall manuscript for language expression, syntax and grammar to the best of my knowledge I believe is correct. I trust that the manuscript has been streamlined from the previous version. Thank you.
All the changes, modifications additions, references, etc, are highlighted in yellow for easy access to the information. The accepted and highlighted manuscript versions in the Zipped folder are the collective final versions of the corrected errors from all reviewers. The files that were not resent from the initial resubmission remain the same.
I would also be grateful to the reviewer, if he/she can please be more specific in his revision request and advise the author, which parts, paragraphs sections etc, suggests to be removed to reduce further the size of this manuscript.
Thank you
Yours sincerely
The author
Kostas A. Triantaphyllopoulos
Reviewer 2 Report (New Reviewer)
Comments and Suggestions for Authors
Triantaphyllopoulos provides a succinct review of lncRNAs and their function in IBD and Johne’s disease. The author provides a detailed account of inflammatory bowel disease and Johne’s disease followed by the description of lncRNAs and their contribution to these diseases.
I believe that the author provides an organized review of lncRNA-mediated regulation of IBD and Johne’s disease. I believe that the manuscript would be beneficial to the scientists working in the field. However, the following points should be considered prior to warranting publication:
1. Lines 353-355: I think that “pseudogenes” is an old definition of ncRNAs. I would like to propose the use of “processed transcripts” instead.
2. Lines 376-379, I do not know if it would be proper to state “ lncRNAs containing a single exon”… I would like to suggest reconsideration of this term.
3. Lines 434-435: It is OK to state that antisense lncRNAs are transcribed from the opposite strand. But I am not sure if it is suitable to state that they are transcribed from “inside of a protein coding gene”
4. Table 2, 3 and 4 should be reformatted as it is difficult to follow them in their current forms
5. Lines 5-10 (right after Table 3), this sentence has been stated before. I think it can be omitted.
6. The resolution of Figure 2 must be improved.
Minor points:
1. Line 46, “10 000” should be “10,000”. Please also check the use of other numbers throughout the manuscript, including the numbers in Table 1.
2. Lines 48-49, the sentence should be revised
3. Lines 53-54, a more proper citation would be better
4. Lines, 159-168, ncRNA and lncRNA abbreviation should be made once. Repetetive abbreviations should be avoided. Please check the rest of the manuscript as well.
5. Lines 209 and 274, please remove one of the periods at the end of the sentences
6. The statements in lines 403-405 and 406-409 are almost the same. Please revise one of them.
7. Line 451, “lncRNAs are lncRNAs that..” please revise this sentence so as to remove one of the “lncRNAs”
8. Lines 460-466, there are repetitive sentences. Please revise.
9. Lines 471-471, this sentence is vague. Please revise.
10. Lines 477-479, too long of a sentence. Please revise to increase clarity
11. Lines 485-488, please revise this sentence to increase clarity
12. Lines 502-508, this sentence is too long. Please revise
13. Line 539, “for” should be “For”
14. Line 286, please fix “events”
15. Line 382, “TFN- should be “TNF” and also use the italic gamma.
Comments on the Quality of English LanguageEditing by a native speaker would be needed to increase the clarity and cohesion of the Manuscript.
Author Response
Please see the attachment

Reviewer 3 Report (New Reviewer)
Comments and Suggestions for Authors
Review for the manuscript "Long Non-Coding RNAs and Their "Discrete" Contribution in IBD and Johne's Disease. What Stands Out in the Current Picture" submitted to IJMS.
Dear Editor, thank you for the opportunity to review this interesting manuscript. However, I have some suggestions before it can be published.
COMMENTS REGARDING LANGUAGE
I suggest that the author check all the manuscript regarding punctuation and grammar.
TITLE
1- Please do not use a full stop in the title.
2- Should the title be: “Long Non-Coding RNAs and Their Discrete Contribution in IBD and Johne's Disease. What Stands Out in the Current Picture?”
3- I also suggest including in the title that this is a comprehensive review.
ABSTRACT
I suggest that the author check MDPI guidelines for building the Abstract. It should contain only 200 words.
KEYWORDS
There are too many keywords. Please, select some of them.
INTRODUCTION
1- Please, include newer references in this section.
2- In this section, the author says (lines 44-47), "Nowadays, the serious concerns in the disease front, have achieved the publication of more than 10 000 papers annually worldwide, in the field of infectious diseases includ ing livestock and domestic animals. Importantly, in humans this amount increases each year [1]..." However, the reference for this sentence was published in 2016 (1. Ducrot C, Gautret M, Pineau T, Jestin A (2016) Scientific literature on infectious diseases affecting livestock animals, 583 longitudinal worldwide bibliometric analysis. Veterinary research 47:42. doi:10.1186/s13567-015-0280-2). Please cite a more recent reference.
3- In lines 142-152 we read that “In recent years, the incessant microbiological research has attracted the most power ful available resources in combating the life-threatening infectious diseases, rendering this exploration significantly necessary and of high priority to assessing the “quiver” that thepathogens carry and the susceptibility profile of the host. More importantly, new discoveries in the genetic and non-genetic components that have been made in pathologies associated to bacterial invasion in human or animal cell have paved the way to an advanced and in depth learning of the interactions between the molecular entities and environmental factors that are likely leading to IBD-related pathogenicity. The ongoing research is significant and has strengthened the exploration and discovery of potential disease biomarkers to improving diagnostic tools and examine new therapeutic modalities of high priority…” Please include references here.
4- The same comment is for lines 153-158.
5- In lines 183-184 we read “In this review, we will provide information and insight for the discovery and gene 184 regulation by lncRNAs supported by database searches…” As I can see, there is only one author in this manuscript.
6- References are also missing in lines: 338-347; 364-368; 459-463; 467-473; 476-483; 485-491; 549-562.
7- The numbers of the lines in the text should be continuous. However, the numbers started again after the tables. References are missing in lines 12-19; and in many other text parts. Please check and add references.
8- Some parts of Table 3 have too much information. Please, reduce.
9- ¨Table 4 is confusing. It isn't easy to understand the top of it. Please, rebuild it.
10- Figure 2 can be improved. The blue color is not good. Please, choose a ‘softer’ color.
11- ↆ symbol is missing in the legend of Figure 2.
CONCLUSION
1- In this section we can find in lines 523-527: In recent years, a limited number of proteins or peptides encoded by ncRNAs, have been demonstrated to exhibit significant biological and pathological functions in the trigger and progression of intestinal barrier dysfunction and the importance of intestinal microbiota. Numerous studies have shown a link between intestinal dysbiosis and IBD[195,235]”… Please do not include references in the Conclusion section.
2- The same for “…as a signature of the faecal microbiota in patients with CD [236]; more importantly, complications during mycobacterial infections have been proposed as a route for malignant pathogenesis. To expand the latter point further, and beyond the anticipated zoonotic risk that microbes and 540 infectious agents pertain, they are also known causes of human cancers, while colorectal 541 cancer (CRC) is a complication of UC and colonic CD, the two main forms of idiopathic 542 inflammatory bowel disease (IIBD) [63]. Please, remove references 236 and 63. They should be in the Discussion section and not here.
3- The same for …” as recently 552 reported [237], as a secure route for a successful outcome, as inaccuracy or contamination 553 fosters high risk in infectious diseases....” Remove reference 237 form this section.
4- In line 537 there are two full stops in the end of the sentence.
5- There is no need to include the website address in this sentece “…with SARS-CoV2 virus in 2020, influenza pandemic 1918-19, 560 (https://www.cdc.gov)... “
LIMITATIONS
Please, include the limitations of this study.
REFERENCES
1- As pointed out above, please include more references published in 2022 and 2023.
2- Remove references from the Conclusion section.
3- I think that many references are not following MDPI guidelines. Please, check.
Comments on the Quality of English LanguageModerate changes are necessary.
Round 2
Reviewer 1 Report (Previous Reviewer 3)
Comments and Suggestions for Authors
It can be published now
Reviewer 2 Report (New Reviewer)
Comments and Suggestions for Authors
The author has sufficiently improved the manuscript to warrant publication.
Comments on the Quality of English LanguageNone.
Reviewer 3 Report (New Reviewer)
Comments and Suggestions for Authors
Dear authors,
Thank you for performing the corrections.
Minor
This manuscript is a resubmission of an earlier submission. The following is a list of the peer review reports and author responses from that submission.
Round 1
Reviewer 1 Report
Comments and Suggestions for Authors
I find the topic of the review interesting, however, it is extremely long, 118 pages is almost a book. Adding the 10+ figures and 10+ tables makes it uninteresting to read. A lot of work by the author, I think you should consider reducing the length of your article. Generally, a proper extension uses between 90 - 150 references, not 522.
Reviewer 2 Report
Comments and Suggestions for Authors
Reviewer comments and suggestions
The authors in this review discussed the recently characterized, novel as well as uncategorized Long non-coding lncRNAs that have been discovered and share in common both gastrointestinal pathologies IBD and JD. It also includes lncRNA regulation, molecular networks and lncRNA interaction lists, etc. as well as presenting the established and potential novel biomarkers pertinent for the above cattle and human disease phenotypes.
Overall, the manuscript was well written. However, a few concerns/comments needed to be explained/modified.
1. I have found some other subsections that are not relevant to the paper as suggested by the title of your paper.
2. Line 26-30 Please avoid long sentences.
3. Line 33-38 few points are repeated, i think the authors could make it short for the simple understanding of the common readers.
4. I did not find methods for drawing figures in the manuscript.
5. I have seen the tables, a few tables need to be arranged well
6. please check the journal style, its not based on MDPI guidelines.
Reviewer 3 Report
Comments and Suggestions for Authors
I think the work is interesting but enormous, probably suitable for a book chapter, if not a whole book!
Even in the proofreading phase, it is hard to read it in its entirety.
So I think it should be reduced significantly, focusing more on the main topic, while everything that concerns long noncoding RNAs in a particular way should be sent elsewhere.
In addition, the role of diet in expression should be considered more specifically, as this could give false results if they are used as biomarkers.